Resource

# Unbinned contigs expand known diversity in the global microbiome

Vishnu Prasoodanan PK[1,11], Oleksandr M. Maistrenko[2,9,11], Anthony Fullam[3], Daniel R. Mende[4], Ece Kartal[5], Luis Pedro Coelho[6], Anja Spang[2], Peer Bork[3,7,8,10] & Thomas S. B. Schmidt[1,3] ✉

The ongoing census of microbial life is hampered by disparate sampling across Earth's habitats, challenges in isolating uncultivated organisms, limited resolution in taxonomic marker gene amplicons and incomplete recovery of metagenome-assembled genomes. Here we quantify discoverable Bacterial and Archaeal diversity in a comprehensive, curated cross-habitat dataset of 92,187 publicly available metagenomes. Clustering 502 million sequences of 130 marker genes, we predict ~705,000 Bacterial and ~27,000 Archaeal species-level clades, the vast majority of which were hidden among unbinned contigs. We estimate that ten and 145 previously undescribed Archaeal and Bacterial phyla, respectively, are discoverable in this dataset. We identify soils and aquatic environments as hotspots of discoverable lineages, but predict that undescribed taxa remain abundant across all habitats. Finally, we show that prokaryotic diversity appears to arise within common evolutionary patterns, as clade size distributions follow power laws, consistently across the Tree of Life.

Microbial life on Earth has deep evolutionary roots and is ubiquitous and abundant in extant ecosystems: prokaryotes (that is, Bacteria and Archaea) emerged more than 4 billion years ago[1], have been detected in every environment on the planet and are estimated to account for ~$10^{30}$ cells and $10^{16}$–$10^{18}$ g of biomass[2,3]. This long history of evolution across a comprehensive ecological range is reflected in a great accumulated phylogenetic diversity, vastly exceeding that of multicellular life[4]. Yet, it remains unclear just how diverse extant Bacteria and Archaea really are, with extrapolations from abundance distributions of individual samples predicting anywhere between $10^6$ and $10^{12}$ prokaryotic species worldwide[5–8].

Only a fraction of this diversity has been accounted for in existing data. Using 16S ribosomal RNA (rRNA) amplicon sequences, previous studies estimated the number of species-level operational taxonomic units (OTUs) in publicly available datasets to be 35,500 in 2004[9], 210,000 in 2014[8], 109,000 in 2016[10] and 740,000 in 2019[7]. Based on rarefaction curves, tracking the number of newly discovered types (species) as more data (samples) are added to the survey, these analyses concluded that, at least in some habitats, the rate of species discovery was slowing down.

Although 16S rRNA data continue to be more abundant in public repositories[11], integrated surveys of prokaryotic diversity increasingly rely on isolate genomes and metagenome-assembled genomes (MAGs) to overcome some of the limitations of amplicon-based datasets (Supplementary Discussion). In metagenomic data, clade-level diversity is often defined based on sequence similarity in a subset of

[1]APC Microbiome and School of Medicine, University College Cork, Cork, Ireland. [2]Department of Marine Microbiology and Biogeochemistry, Royal Netherlands Institute for Sea Research, Texel, the Netherlands. [3]Molecular Systems Biology Unit, European Molecular Biology Laboratory, Heidelberg, Germany. [4]Human Biology-Microbiome-Quantum Research Center (WPI-Bio2Q), Keio University, Tokyo, Japan. [5]Institute for Computational Biomedicine, Faculty of Medicine, University of Heidelberg, Heidelberg, Germany. [6]Centre for Microbiome Research, School of Biomedical Sciences, Queensland University of Technology, Translational Research Institute, Brisbane, Queensland, Australia. [7]Max Delbrück Center for Molecular Medicine, Berlin, Germany. [8]Department of Bioinformatics, Biocenter, University of Würzburg, Würzburg, Germany. [9]Present address: Swammerdam Institute for Life Sciences, University of Amsterdam, Amsterdam, the Netherlands. [10]Deceased: Peer Bork. [11]These authors contributed equally: Vishnu Prasoodanan PK, Oleksandr M. Maistrenko. ✉e-mail: sebastian.schmidt@ucc.ie

near-universal taxonomic marker genes[12–14] or entire genomes[15]. For example, proGenomes3 (ref. 16) encompasses 41,171 species-level clusters of high-quality isolate genomes from the National Center for Biotechnology Information's RefSeq and GenBank databases, whereas the Genome Taxonomy Database (GTDB) r226 delineates 143,614 species, combining isolate genomes and high-quality MAGs[17]. In addition, large-scale MAG catalogues continue to add to the census, reporting for example 4,644 species-level genomes in the human gut microbiome[18], 8,300 species in the ocean[19], 21,077 species in soils[20], between 100 and 13,000 species for the various habitats in the MGnify Genomes collection[21] or 18,000 species across the habitats represented in the IMG/M database[22]. We recently developed SPIRE[23], which provides an integrated survey of microbial diversity across 99,146 manually annotated metagenomes from all sampled habitats on Earth, describing 107,068 species-level clusters, of which 92,134 are entirely based on MAGs (that is, lacking a cultivated representative).

Such genome-centric surveys have greatly expanded the scope of study for microbial diversity, ecology and evolution, in particular for uncultivated taxa: in GTDB r226, just 64 out of 189 phyla and 34,500 out of 143,614 species contain cultivated representatives. In other words, 66% of currently recognized phyla and 76% of species have been described solely based on MAGs. Nevertheless, MAG recovery remains limited in terms of both precision (due to residual binning artefacts[24]) and recall (most bins remain incomplete with current automated binning tools[25]). Indeed, with common workflows, most metagenomically assembled contigs remain unbinned: in SPIRE, just 2.36 terabase pairs (or 10%) of the total assembly are assigned to medium- or high-quality MAGs. These unbinned contigs circumscribe a space of discoverable microbial diversity (that is, the reservoir of recoverable genomes that remains untapped by current workflows). Recent studies estimated this genomically unrepresented diversity at around 135,000 species in 18,000 metagenomes[26] and 83,000 species in 249,000 metagenomes[27]. Novel tools have emerged to make these genomes accessible: the recently developed Bin Chicken workflow of targeted co-assembly recovered genomes for 24,028 previously unrepresented species from just 800 sample groups[28].

In compiling this Resource, we conducted a survey of discoverable Bacterial and Archaeal diversity among a set of 92,187 metagenomes, sampled across Earth's microbial habitats. We tracked the diversity represented in 120 Bacterial[13] and 53 Archaeal[14] marker genes originating from reference isolate genomes[16], MAGs[17,23] and unbinned metagenomic contigs to quantify the number of lineages to be incrementally discovered in each set. Using habitat-stratified rarefactions, we tallied the amount of diversity that remains to be discovered in available sequence data and estimated the additional diversity expected to emerge as new metagenomes are added to the survey. We tracked lineages across marker gene trees to quantify how many deeper clades (that is, unrecognized phyla, classes, orders, families and genera) remain undiscovered across different habitats. We show that the size distributions of Bacterial and Archaeal clades (including those inferred from unbinned genes) follow power laws, in line with century-old hypotheses by Willis[29] and Yule[30], with ramifications for biodiversity theory and the study of microbial evolution. In other words, we ask how many and which microbial lineages are hiding in plain sight among unbinned contigs because they are missed by current toolkits, and how many more lineages we are poised to discover as we continue to sample Earth's habitats.

## Results

### Just 20–50% of discoverable species are captured by genomes

We tracked the incremental discovery of Bacterial and Archaeal diversity among 92,187 metagenomic samples in SPIRE[23] along rarefaction curves using species-level clusters of assembled taxonomic marker genes (Fig. 1a). We estimated the total discoverable diversity in the studied dataset to be ~705,000 Bacterial and ~27,000 Archaeal species (Supplementary Table 1 and Fig. 1b); rarefaction trajectories (Extended Data Fig. 1) and overall species count estimates (Extended Data Fig. 2) were remarkably consistent across the different independently queried marker genes. Just ~20,000 Bacterial and ~800 Archaeal species comprised cultivated representatives in proGenomes3; an additional ~25,000 and ~1,700 species (for a total representation of 6.4 and 8.8%, respectively) mapped to GTDB r220; and a further ~80,500 and ~4,200 species were contributed by SPIRE MAGs, for a total species-level diversity representation within genome-based datasets of 17.8% among Bacteria and 24.6% among Archaea. In other words, up to ~75–80% of species-level clusters were not captured by genomes of cultivated organisms or MAGs.

These unbinned species were over-represented among uncultivated phyla (recognized solely based on MAGs in the GTDB, with no cultivated isolates; Cohen's $d = 0.33$; Wilcoxon $P = 0.01$; Extended Data Fig. 3a) and, compared with species with a cultivated representative, were associated with higher genomic GC content ($R^2 = 0.15$; $P = 1.6 \times 10^{-16}$), but not average genome size ($P = 0.338$), coding density ($P = 0.418$) or estimated clade size ($P = 0.889$) in a multiple linear regression (Extended Data Fig. 3b). Species-level cluster sizes followed a long-tailed distribution, driven by a large fraction of singleton clusters (containing just one sequence; Extended Data Fig. 4). Although this is consistent with expectations of the so-called rare biosphere of lineages with low prevalence and abundance[31], a subset of singleton clusters may also arise from assembly artefacts or spurious sequences. When conservatively considering only non-singleton marker gene clusters (containing ≥2 sequences from independent sources), diversity estimates were accordingly reduced to a total of ~249,000 Bacterial and ~12,800 Archaeal species (Fig. 1a,b and Supplementary Table 1); this can be considered a lower bound (Supplementary Discussion), yet still corresponded to an increase of 98% relative to the discoverable species set represented by genomes. Indeed, although around three in four unbinned marker gene clusters were singletons, these accounted for <20% of unbinned genes, with the remainder mapping to reference genomes, MAGs or larger unbinned gene clusters (Extended Data Fig. 4). Taken together, this suggests that the vast majority of unbinned genes represent a real biological signal and probably belong to low-quality (incomplete or contaminated) MAGs, enriched in uncultivated clades.

At the same time, marker genes for around half of the species defined by (isolate) genomes in proGenomes3 and GTDB r220 were not represented in metagenomic assemblies (total reference dataset sizes indicated by dotted lines in Fig. 1a), and around 40% were singletons (Extended Data Fig. 4). This suggests that species with cultivated representatives and those pinpointed for isolation are themselves often rare, occurring at low prevalence and/or low abundance in environmental data. Thus, the discovery gap goes both ways: isolate genomes and MAGs captured only 20–50% of discoverable species in metagenomic contigs, but metagenomic assemblies likewise failed to capture around half of the diversity represented by isolates.

### Species discovery remains in full swing across Earth's habitats

The number of discoverable species varied greatly between individual habitats (Fig. 1b), with moderate correlation to sampling effort ($\rho_{Spearman} = 0.51$ for Bacteria and 0.09 for Archaea). Estimated Archaeal species counts were generally two to three orders of magnitude below those for Bacteria in host-associated and anthropogenic habitats (including some environments with almost no detectable Archaea), roughly within 1 log in aquatic habitats and soils and almost on par in extreme environments. Genomic representation of discoverable species likewise varied across habitats, but was generally within one order of magnitude of the total.

We next tested whether species discovery showed any signs of slowing down by calculating species discovery coefficients (termed $\alpha$) based on individual, habitat-stratified rarefaction curves (Fig. 1c and

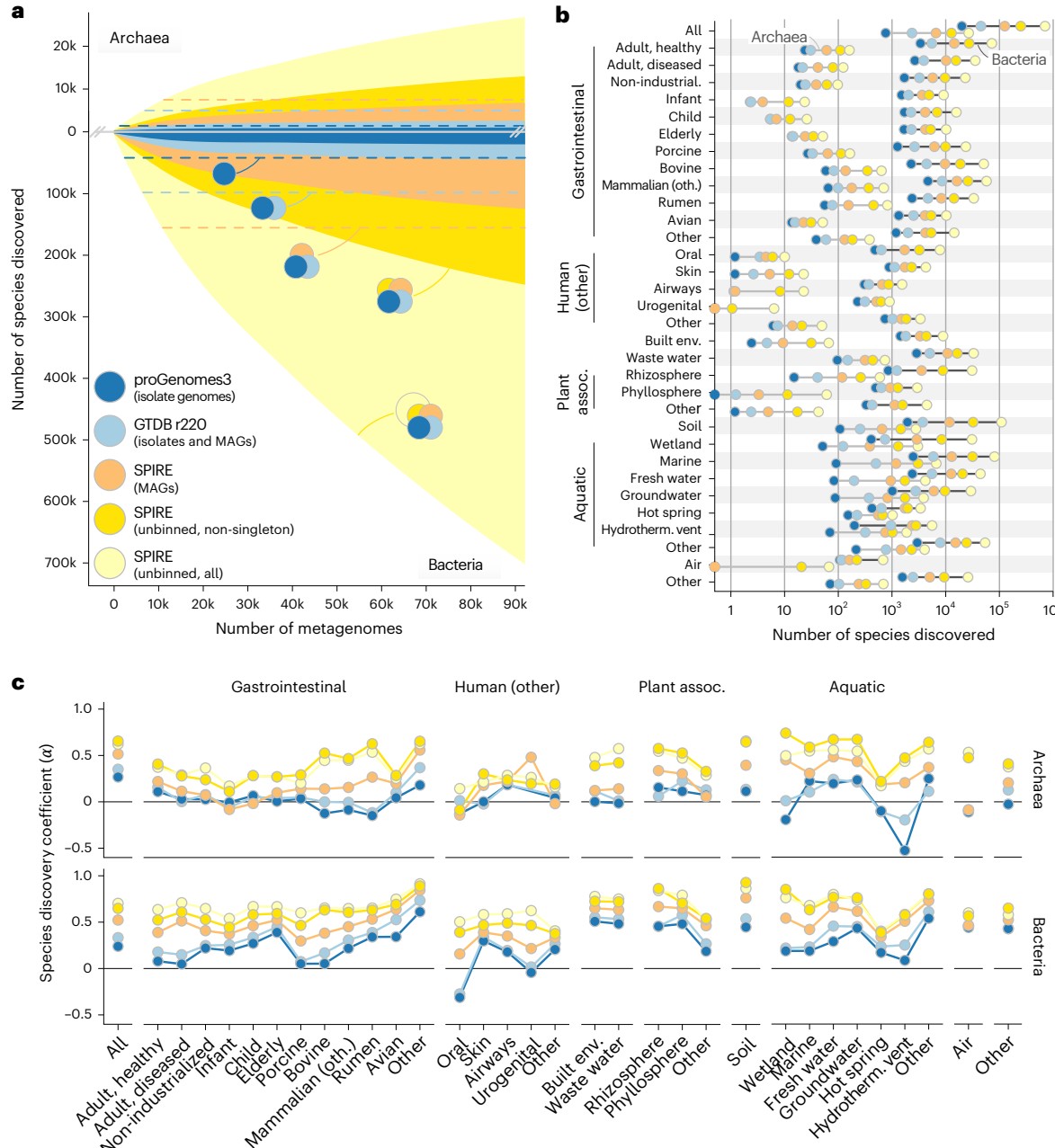

**Fig. 1 | Extensive discoverable diversity among unbinned contigs.**
**a**, Rarefaction curves of Archaeal (top) and Bacterial (bottom) species discovery across ~90,000 metagenomes. Based on marker gene cluster composition, we distinguished five hierarchically overlapping categories of data provenance for every inferred species (indicated as Venn diagrams): (1) those including an isolate genome represented in proGenomes3 (dark blue data series); (2) additional species (beyond proGenomes3) represented in GTDB r220 isolate genomes and MAGs (light blue); (3) species defined from SPIRE MAGs (orange); and species-level marker gene clusters exclusively based on unbinned contigs (that is, not assigned to medium- or high-quality MAGs), further distinguishing (4) non-singleton clusters (containing ≥2 sequences from independent sources; dark yellow) among (5) all marker gene clusters (light yellow). Reference levels are indicated as dashed lines (that is, the total number of species in proGenomes3, GTDB r220 and SPIRE

MAGs, including those not discoverable in metagenomes). **b**, Total number of species-equivalent marker gene clusters per habitat (Supplementary Figs. 1 and 2 provide corresponding habitat-stratified rarefaction curves). **c**, Species discovery coefficients ($\alpha$), calculated from habitat-stratified rarefaction curves (Methods). Values for $\alpha$ in [0, 1] correspond to unsaturated species discovery curves where additional samples continue to add species to the survey (analogous to open pangenomes). Lower $\alpha$ values indicate a more pronounced flattening off in the rarefaction curve, indicating a more pronounced slowdown in species discovery. Higher $\alpha$ values indicate a less pronounced decrease in the rate of species discovery. For $\alpha \rightarrow 1$, species discovery is fully unsaturated, meaning that each newly added sample adds species to the survey, with no discernible flattening off in the species discovery curve. assoc., associated; env., environment; hydrotherm., hydrothermal; industrial., industrialized; k, thousand; oth., other.

Methods). We observed that genome-based discovery of Archaeal species was fully saturated ($\alpha \le 0$) or approaching saturation (low positive $\alpha$) in most host-associated and anthropogenic habitats. Specifically, all Archaeal species represented in proGenomes3 and GTDB r220 that can be metagenomically discovered using current approaches have

already been accounted for in habitats such as human infant gut, oral or skin samples; adding more samples to the survey is not expected to increase metagenomic representation. Similarly, anthropogenic environments (built environments and wastewater), wetlands and extreme environments (hot springs and hydrothermal vents) appear

saturated in terms of the metagenomic discovery of Archaeal reference species, whereas even MAG-based discovery showed clear signs of slowing down. In contrast, Archaeal species discovery coefficients among unbinned contigs were substantially higher, indicating that the existing large discovery gap between binned and unbinned contigs will continue to widen.

Bacterial species discovery coefficients were generally higher, indicating that, with current approaches, Bacterial diversity will continue to be discovered at higher rates than for Archaea. The human mouth, urogenital tract and adult gut, as well as porcine and bovine gut environments, are the only tested habitats in which the metagenomic discovery of isolate-represented diversity is approaching saturation. Yet, even in these habitats, as across all others, MAG-based species discovery is expected to continue at high rates, outpaced only by species discovery among unbinned contigs ($\alpha \geq 0.5$ across most habitats). Soils, wetlands, the rhizosphere, freshwater habitats and the gastrointestinal tracts of non-mammalian, non-avian animals stood out as particular hotspots of untapped diversity, with virtually no slowdown in species discovery ($\alpha \geq 0.8$). Overall, the marked difference in species discovery coefficients between reference genome sets, MAGs and unbinned contigs suggests a widening gap between discovered (genomically represented) and discoverable (unbinned) diversity, as the latter will continue to outpace the former.

Beyond species discovery within individual habitats, we explored how much each habitat contributed to the total survey. We tracked habitat-stratified species accumulation curves, quantifying how much diversity each sequentially added habitat contributed incrementally. This revealed striking discrepancies between sampling effort and observed unique diversity (Extended Data Fig. 5). Although human-associated, gastrointestinal and built-environment metagenomes represented almost three-quarters of sampling effort, they only accounted for 38 and 8% of discoverable Bacterial and Archaeal species, respectively. In contrast, soils and aquatic habitats—in particular wetlands—remain major reservoirs of unexplored microbial diversity (Supplementary Discussion).

### Uncaptured deeply branching lineages may enrich the Tree of Life

We next explored how these trends translate to deeper taxonomic levels. We inferred gene phylogenies for each of the considered 53 Archaeal and 122 Bacterial markers, estimated the number of represented genus-, family-, order-, class- and phylum-level clades by cutting each tree at level-specific relative evolutionary divergence (RED)[32] cutoffs, and summarized the resulting clade counts (Methods). As shown in Fig. 2 for Archaeal marker RNA polymerase subunit E (*rpoE*), unbinned sequences added considerable tip-level (that is, approximately species-level) diversity for most recognized clades and often also provided relevant ecological context. For example, for the family Nitrosopumilaceae (within the Thermoproteota phylum[33]) that encompasses several known ammonia-oxidizing lineages, we observed a substantial expansion of covered diversity among marine genera (for example, Nitrosopelagicus or *Nitrosopumilus*) while linking other subclades to wetland-, groundwater-, soil- and plant-associated habitats, supported across several phylogenies.

Marker gene phylogenies recovered previously recognized deep-branching clades as mostly homogenous groups and recapitulated common hypotheses regarding their branching order[34]. Our estimates of reference phylum-level clades (containing sequences from proGenomes3 or GTDB) exceeded the number of phyla recognized in GTDB r220 based on full species trees for both Archaea (estimate: $36 \pm 12$; GTDB r220: 19) and Bacteria ($238 \pm 82$; GTDB: 175), in part because our RED partitioning algorithm tended to split large clades (such as the TACK superphylum, comprising Thaumarchaeota (Nitrososphaerota), Aigarchaeota, Crenarchaeota and Korarchaeota; represented as the single phylum Thermoproteota in the GTDB) or

groups predicted to have skewed evolutionary rates (such as DPANN Archaea; named for the phyla Diapherotrites, Parvarchaeota, Aenigmarchaeota, Nanoarchaeota and Nanohaloarchaeota) into multiple phyla. All explored phylogenies contained varying numbers of deep (phylum-level) branches encompassing only sequences in unbinned contigs or SPIRE MAGs. For example, the Archaeal *rpoE* tree in Fig. 2 contains an unclassified phylum-level branch comprising 758 sequences from different habitats, organized into 109 tips (gene clusters) and five predicted family-level clades. Overall, we estimated a median of ten putative phylum-level clades in Archaea ($4 \pm 3$ from SPIRE MAGs and $6 \pm 8$ among unbinned contigs), corresponding to an increase of ~28% over estimated reference clades, and 145 clades in Bacteria ($28 \pm 29$ among MAGs and $117 \pm 137$ in unbinned contigs), corresponding to an increase over reference estimates of ~61%. Towards shallower taxonomic levels, our estimates of reference clade counts approximated those in GTDB r220 (with the exception of the species level; see above), whereas estimates on discoverable clades among SPIRE MAGs and unbinned contigs strongly increased with increasing taxonomic resolution (Fig. 3a and Table 1).

The recovery of predicted clades was heterogeneous across habitats (Fig. 3b). Gut and non-intestinal human samples covered only a small fraction of predicted clades, at both deep (phylum-level) and shallower (genus-level) resolution, despite comprising the majority of the sequencing effort. The majority of predicted higher-level clades were observable in soil, rhizosphere, wastewater and aquatic environments. Hydrothermal vents stood out as particular hotspots of Archaeal diversity: accounting for just 296 metagenomes in our dataset (0.3% of the total), they contained representatives of four-fifths of reference and two-thirds of MAG-based and unbinned predicted Archaeal phylum-level clades, representing the widest coverage of deep lineages observed for any tested habitat.

### Prokaryotic diversity follows Willis' law and Yule curves

In 1922, Willis observed that taxonomic clade size distributions of several plant and animal groups follow power laws: the frequency of genera decreases with the exponent of genus size[29], resulting in hollow curves dominated by singleton clades (containing just one species), with a heavy tail of very large clades (disproportionally species-rich genera). Beginning with Yule's seminal work explaining this observation by postulating preferential attachment of novel species to existing genera, proportionally to genus size[30], researchers have sought to identify the evolutionary mechanisms and processes that give rise to extant biodiversity[35–39].

We hypothesized that prokaryotic diversity follows the same empirical laws and reasoned that our dataset was uniquely suited to test this, as our clade-level groups were data driven (inferred based on RED values), taxonomy agnostic (defined from marker phylogenies) and comprehensive (sampled across Earth's habitats). We found that Archaeal and Bacterial diversity indeed follow Willis' law: clade frequencies decrease with increasing clade size in a power law relationship (Fig. 4), along the full tested range of more than five orders of magnitude of clade counts. Willis' law held across taxonomic scales, from species within genera to species within phyla (Fig. 4a), and also for subclades at higher taxonomic levels, from genera within families to classes within phyla (Fig. 4b). Fitting naive power law equations for each individual marker gene, we computed a marker-gene-specific Willis coefficient $\omega$ from the slope in a log–log plot (Methods). We observed a clear trend of decreasing $\omega$ (implying a heavier tail; that is, stronger bias towards large clades) across taxonomic levels for species counts (from a $\omega$ value of ~1.4 for species within genera to a $\omega$ value of ~0.1 for species within phyla), and more similar $\omega$ values ranging from 1.4–1.7 for subclade-within-clade counts (Extended Data Fig. 6a).

Although naive power laws provided good overall fits, we observed that clade size distributions were disproportionally heavy tailed on the right (large clades were over-represented) while also deviating towards

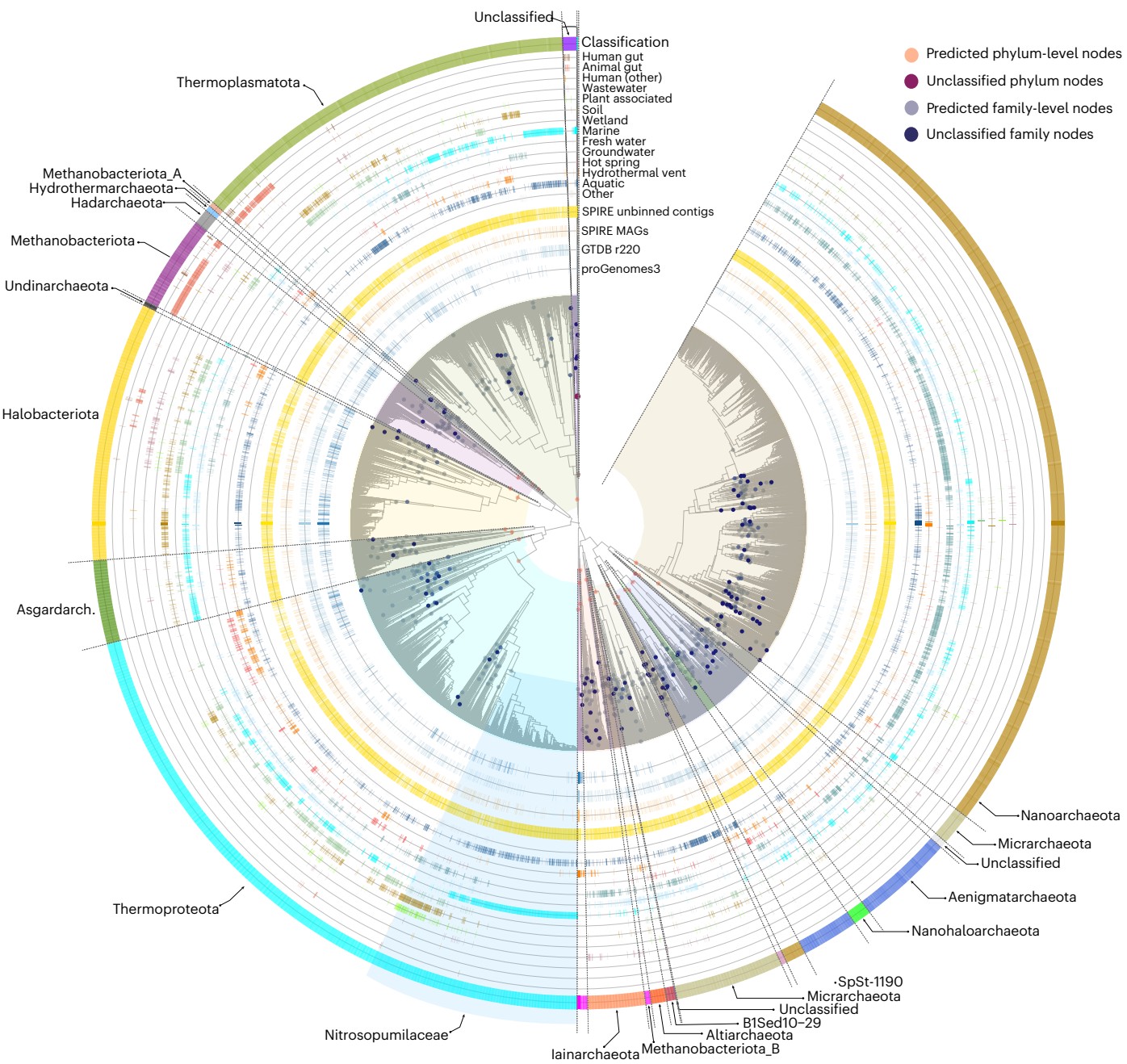

**Fig. 2 | Unbinned contigs enrich the Tree of Life among both known and deep clades.** Example phylogeny of the Archaeal RNA polymerase E gene (*rpoE*; TIGR00448), inferred from pre-clustered reference and metagenomic sequences (Methods). Sequence clusters at the tips are annotated according to their source (proGenomes3, GTDB r220 or SPIRE) and the habitat categories from which they were recovered among the full set of metagenomes. Nodes marked with dots indicate clade-level groups at the phylum (red) and family (blue) level, inferred based on pre-calibrated cutoffs in RED (Methods). Phylum groups were taxonomically classified based on encompassed sequences from proGenomes3, GTDB r220 or SPIRE MAGs. Nitrosopumilaceae are indicated as an example clade where unbinned contigs substantially enrich known taxa. Deep unclassified phylum-level clades are highlighted at various points. Phylogeny visualizations for other Archaeal marker genes are available under European Bioinformatics Institute BioStudies accession code S-BSST2111 (ref. [80]). Asgardarch., Asgardarchaeota.

the left (lower frequencies for small clades), thereby showing characteristics originally used by Yule to deduce underlying processes[30]. We therefore fit the data to Yule–Simon distributions (Fig. 4a,b). Estimates of the Yule–Simon parameter $\rho$ were consistent across marker genes and between Bacteria and Archaea (Extended Data Fig. 6b): $\rho$ decreased with increasing taxonomic depth for species counts ($\rho$ = ~1.0 for genera to $\rho$ = ~0.4 for phyla), with an inverse trend for subclade counts ($\rho$ = ~0.85 for genera within families to $\rho$ = ~1.50 for classes within phyla). $\rho$ can be thought of as a rich-get-richer coefficient that describes preferential attachment in a Yule process: low $\rho$ values indicate a strong preferential attachment of new subclades to large existing clades. Therefore, our results indicate that new species tend to arise in larger existing clades, with a skew that is strongest in phyla and weakest for genera (that is, the dominance of species-rich phyla, such as Bacillota or Pseudomonadota, is more pronounced than that of species-rich genera, such as Prochlorococcus). The inverse was observed for subclades within clades: new genera arise with a stronger bias towards existing large families than new classes towards existing large phyla. For orders

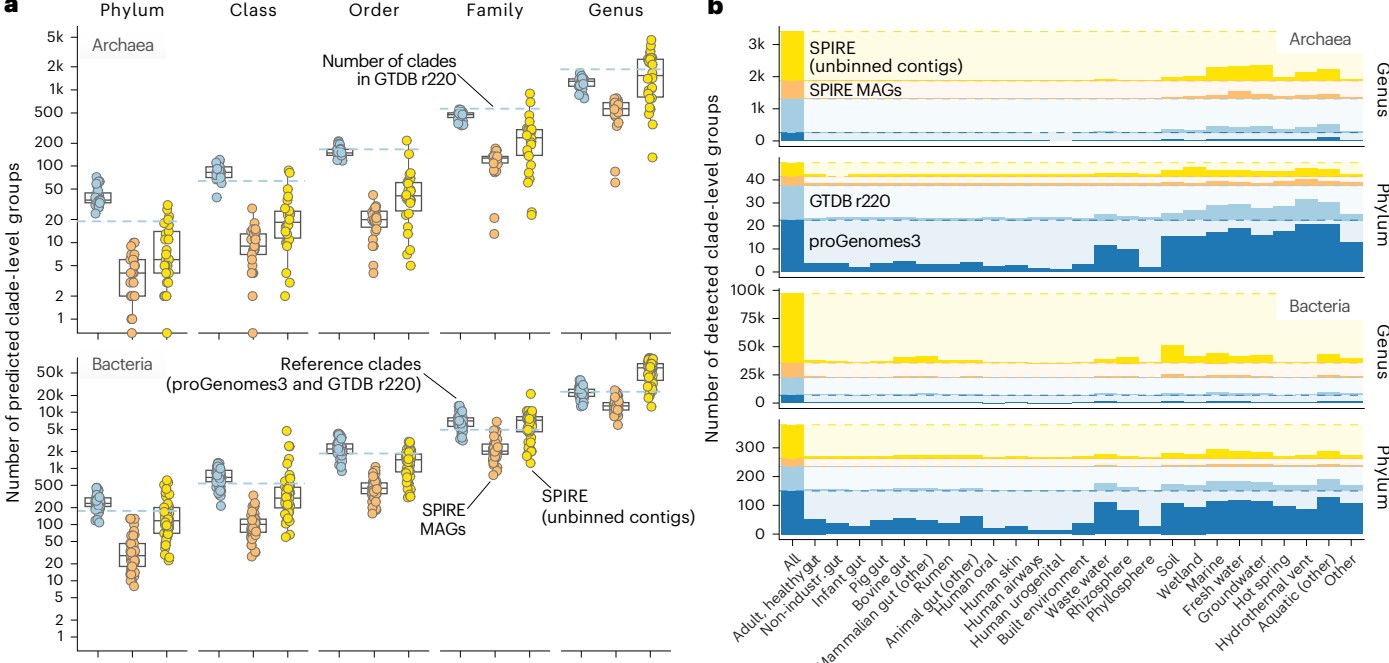

**Fig. 3 | Unbinned contigs and MAGs suggest thousands of discoverable deeper clades. a**, Numbers of predicted clade-level groups at the phylum, class, order, family and genus level, based on RED cutoffs. Each point indicates an estimate for one marker gene ($n = 29$ for Archaea and $n = 53$ for Bacteria; Methods). Colours indicate data source: blue for clades containing at least one reference sequence (from proGenomes3 or GTDB r220); orange for those containing no reference sequences, but at least one SPIRE MAG; and yellow for clades containing only sequences from unbinned SPIRE contigs. The dashed blue lines indicate the reference number of clades per level in GTDB r220. The boxplots indicate median values, 25th and 75th percentiles (box edges) and 1.5× the interquartile ranges (whiskers). **b**, Distribution of discoverable phylum- and genus-level clades across habitats. The bars indicate how many clades for each data source are discoverable across all habitats (leftmost column) or when only considering data from individual habitats.

## Table 1 | Estimated clade counts at deeper taxonomic levels

| Database | Phylum | Class | Order | Family | Genus | Species |
|---|---|---|---|---|---|---|
| **Archaea** | | | | | | |
| GTDB r220 (reference) | 19 | 64 | 166 | 564 | 1,847 | 5,869 |
| GTDB r220 | 36±12.2 | 83±19.7 | 149±25.4 | 472±58.4 | 1,288±215.4 | 2,159±746 |
| SPIRE MAGs | 4±2.9 | 9±5.7 | 20±8.4 | 128±36.5 | 564±187.2 | 4,222±688 |
| SPIRE unbinned | 6±8.3 | 18±21.1 | 41±44.7 | 236±193.6 | 1,536±1,141.4 | 21,210±26,244 |
| **Bacteria** | | | | | | |
| GTDB r220 (reference) | 175 | 538 | 1,840 | 4,870 | 23,112 | 107,235 |
| GTDB r220 | 238±82.1 | 691±251.7 | 2,215±814.1 | 6,999±2,307.7 | 22,347±5,431.8 | 45,147±6,892 |
| SPIRE MAGs | 28±29.3 | 101±56.5 | 443±195.1 | 2,027±1,107.5 | 12,932±3,854.3 | 80,526±7,958 |
| SPIRE unbinned | 117±137 | 296±754.2 | 1,423±672.2 | 7,179±3,367.8 | 61,497±22,186.2 | 580,868±321,417 |

Clade counts were estimated based on RED cutoffs from individual marker gene phylogenies (Methods). The values shown are the median±s.d. across domain-specific marker gene sets. Note that the discrepancy between reference and estimated species counts is in line with observations that around half of marker gene clusters representing reference species were not observed in metagenomic assemblies (main text and Extended Data Fig. 3).

within classes and classes within phyla, we observed a disparity between Archaea and Bacteria that was only in part attributable to sampling noise (fewer Archaeal clades leading to noisier fits). This may indicate that an Archaeal order or class is not fully equivalent to Bacterial orders or classes in terms of phylogenetic depth, or that Archaeal and Bacterial diversity at these levels is organized differently.

### Diversity patterns vary across habitats and phylogeny

We explored how clade size distributions are shaped by environment and phylogeny by refitting curves for individual habitats (Fig. 4c) and phyla (Fig. 4d). Yule–Simon coefficients were generally lower in well-sampled habitats ($\rho_{Spearman} = -0.50$ for Bacteria and $-0.25$ for Archaea), with clear differences between broader environments (Fig. 4c). Intestinal and human non-intestinal habitats showed the lowest $\rho$ values, arguably because they are narrowly defined and correspond to just one or few host species. In contrast, higher coefficients were observed for aquatic environments (in particular hot springs and groundwater) and soils, but also non-mammalian guts, indicating that diversity is less dominated by few large clades and that newly discovered species in these environments are more likely to belong to undescribed deeper clades, whereas they are expected to belong to (known) large clades in the intestine.

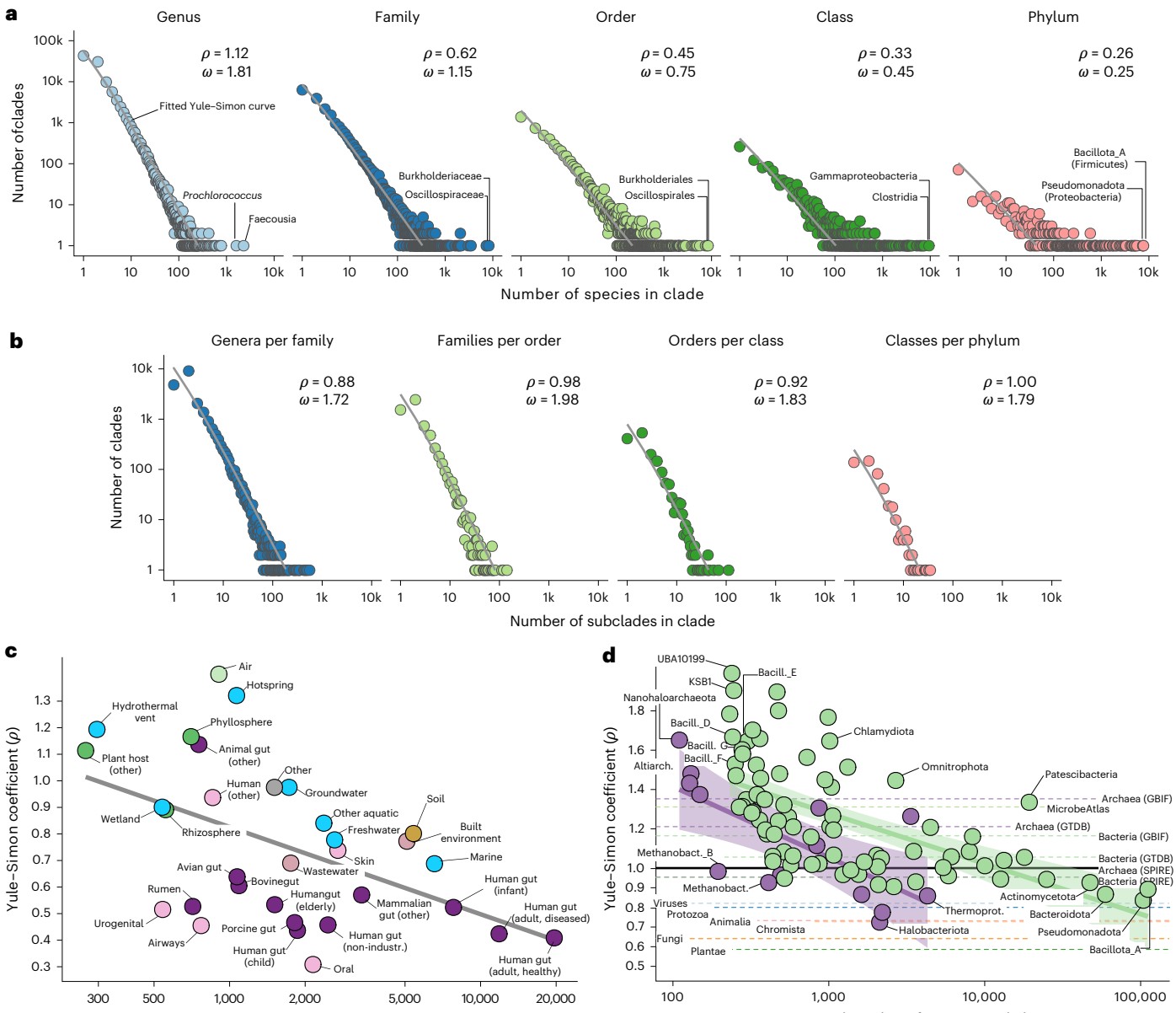

**Fig. 4 | Prokaryotic diversity follows Willis' law and Yule curves. a**, Log–log plots of clade size (number of species per clade; *x* axis) against clade count (number of clades containing *x* species; *y* axis) for the example Bacterial marker gene *rsmD* (16S rRNA guanine methyltransferase; TIGR00095). Clade-level groups were inferred based on RED cutoffs from a full phylogeny at each taxonomic level (Methods). The grey lines indicate fitted Yule–Simon curves. The estimated Willis coefficient ($\omega$) and Yule–Simon coefficient ($\rho$) are provided (Methods). **b**, Equivalent to **a**, but showing the number of clades (*y* axis) containing *x* subclades (*x* axis) of the subordinate taxonomic level (that is, genera within families through to classes within phylum). **c**, Yule–Simon coefficients estimated for individual habitats (that is, only considering genes assembled from metagenomes of a focal habitat). Grey line indicates a linear fit. The *x* axis corresponds to the number of samples per habitat; colours indicate habitat

categories (purple, gastrointestinal; light pink, other human body sites; blue, aquatic; green, plant associated; dark pink, anthropogenic; brown, soil; light green, air). **d**, Yule–Simon coefficients estimated within individual recognized phyla (Methods). The dotted lines indicate $\rho$ estimated for prokaryotic domains (green, Bacteria; purple, Archaea), eukaryotic kingdoms (dark green, Plantae; orange, Fungi; light orange, Chromista; pink, Animalia; dark blue, Protozoa) and viruses (light green) based on reference taxonomies from the GBIF, GTDB r226 and Microbe Atlas Project 16S rRNA OTUs (Methods). The shaded bands indicate 95% confidence intervals of linear fits. Corresponding Yule curves, analogous to **a**, are shown in Extended Data Fig. 7. Altiarch., Altiarchaeota; GBIF, Global Biodiversity Information Facility; Bacill., Bacillota; Methanobact., Methanobacteriota; Thermoprot., Thermoproteota.

Yule–Simon coefficients were strongly associated with total sampled species richness within recognized phyla for both Bacteria ($\rho_{Spearman} = -0.71$) and Archaea ($\rho_{Spearman} = -0.79$; Fig. 4d): well-sampled phyla such as Pseudomonadota or Bacillota_A had low $\rho$ coefficients, whereas smaller phyla were less dominated by large clades. Two notable exceptions were Methanobacteriota, which deviated towards lower $\rho$ (fewer species sampled, but highly concentrated),

and Patescibacteria, with an estimated ~19,500 species distributed into more evenly sized clades ($\rho = 1.33$). Yet, nearly all tested prokaryotic phyla had higher Yule–Simon coefficients than eukaryotic kingdoms and even viruses, inferred based on reference taxonomies of the Global Biodiversity Information Facility (GBIF; Extended Data Fig. 7). Moreover, domain-wide $\rho$ estimates based on our dataset were substantially lower than those based on GBIF Bacterial and Archaeal taxonomies,

GTDB r226 reference taxonomies[40] or 16S rRNA OTUs from the Microbe Atlas Project[11,41]. This suggests that although eukaryotic diversity (in particular among plants and fungi) is more concentrated into few large clades than for Bacteria and Archaea, the organization of biodiversity follows similar and consistent patterns across the entire Tree of Life.

## Discussion

There are multiple layers of discovery in the description of prokaryotic lineages: the isolation of organisms for experimental access and formal taxonomic approval remains standard[42–45], yet the delineation of genomes directly from metagenomes has enabled the characterization of a greatly extended range of clades without cultivated representatives, and even variants in (amplified) marker gene sequences can be informative of further diversity. In this context, our study addresses four basic questions: (1) how many lineages can be discovered in currently available data; (2) how much of this diversity is missed by current genome-centric approaches; (3) should we expect discovery to slow down as more data come in; and (4) how is prokaryotic diversity organized across habitats and phylogeny?

Our results extend previous observations[18,20,23,46] that isolates greatly underestimate tractable genomic diversity in metagenomes. Yet, strikingly, this discovery gap goes both ways: only half of the species encompassed by genomes of cultivated representatives (and, by extension, reference MAGs) were detected in our metagenomic dataset. These levels of detection are higher than previous reports[26], probably because the present dataset is more comprehensive, although saturating discovery curves for several environments indicate that further detections are not expected when adding more samples (at least when assuming consistent sampling bias). This tallies with the idea that cultivation efforts and metagenomic sampling often target different (micro-)environments and that cultivability does not necessarily reflect abundance in natural communities[47].

The vast majority of discoverable diversity in our dataset extended beyond all genome-centric approaches. As a lower bound, defined as independent detection in at least two distinct samples, one species remains unbinned for every species discovered among cultivated organisms (proGenomes3), reference MAGs (GTDB r220) or dataset-specific MAGs (SPIRE). When considering all detected sequence variants, estimates go up to four discoverable species per genomically defined species, or 580,000 uncaptured Bacterial species and 21,000 uncaptured Archaeal species, exceeding recent reports of 83,000 species based on raw read mapping to conserved marker gene windows using SingleM and sandpiper[27] or 135,000 species based on metagenomic assembly[26].

At broader taxonomic levels, our data likewise suggest a wealth of discoverable clades: we estimate that fewer than one in four Bacterial genera and just three in five Bacterial phyla that can be inferred from assemblies are currently recognized in reference databases. These are estimates across many phylogenies so that issues affecting individual gene trees (such as paralogues or uneven evolutionary rates[48–50]) are expected to mostly even out. Nevertheless, the proper characterization of lineages requires genomic context, or at least phylogenies of multiple concatenated informative markers combined with complex models of evolution[51,52], beyond the scope of the present study. Although these numbers carry increasing uncertainties at broader taxonomic levels, this puts the total (currently discoverable) Bacterial phyla in the range of 300–400, compared with 30–40 among Archaea, although we note that RED-based clade definitions at phylum and class levels were broader in Archaea than in Bacteria. Indeed, this ratio of roughly one order of magnitude difference between Bacterial and Archaeal clade counts held across taxonomic levels, supporting previous observations that Archaea are much less diverse than Bacteria[7,53,54].

We found that Bacterial and Archaeal diversity are organized along common fundamental patterns. Empirical scaling laws are prevalent in microbiology, from biogeographic taxa–area relationships[55,56] to macroecological patterns[6]. Our work provides comprehensive evidence that prokaryotic clade size distributions likewise follow power laws, supporting century-old hypotheses by Willis[29] and Yule[30]. Power laws are a hallmark of fractal geometries, and the remarkable consistency of inferred coefficients—not only between Bacteria and Archaea, but also across eukaryotic kingdoms and viruses, and even for subclades within clades at different depths—implies that consistent (and possibly predictable) evolutionary forces mould the Tree of Life into recurring patterns from root(s) to tips (that is, biodiversity is indeed fundamentally fractal)[35,57–59]. Our data strongly support Yule's hypothesis[30,39] that one central pattern is preferential attachment: the rich get richer as new clades arise preferentially within already large clades. The effect varies across habitats and phylogeny and can also be interpreted in light of clade discovery: assuming random sampling, far more species need to be discovered in intestinal environments or species-rich phyla such as Pseudomonadota (with low Yule–Simon coefficients) than in hot springs or among Patescibacteria (higher coefficients) to identify a previously undescribed deeper clade.

We caution against extrapolating total richness estimates from our rarefaction curves, as future data may break out from previous sampling bias (Supplementary Discussion). Indeed, our inferred rarefaction curves and species discovery coefficients are conservative lower-bound estimates, as they assume continued sampling from the same underlying distribution, whereas in fact the vast majority of the planet remains a metagenomic terra incognita, and overlooked endemic clades in unsampled sites may disproportionately contribute to global biodiversity[60,61]. Despite these caveats, soils, aquatic habitats and non-mammalian guts stood out as key reservoirs of discoverable diversity and hotspots primed for future discovery, although our findings suggest that the discovery of undescribed diversity will continue unmitigated for almost all considered habitats as data continue to accrue near-exponentially in public repositories. In the meantime, even existing data may still hold countless surprises, as vast prokaryotic diversity remains hidden in plain sight.

## Methods

### Data sources and pre-processing

We used three principal data sources in our analysis: (1) proGenomes3 (ref. [16]), a quality-filtered subset of 907,388 isolate genomes sourced from the National Center for Biotechnology Information's RefSeq and GenBank databases[62], organized into 41,171 species-level clusters based on a set of 40 SpecI marker genes[12]; (2) GTDB r220 (April 2024[17]), a curated set of 596,859 isolate genomes and MAGs, clustered into 107,235 Bacterial and 5,869 Archaeal species-level clusters sharing ≥95% whole-genome average nucleotide identity; and (3) SPIRE version 1.1 (ref. [23]), encompassing a curated set of 92,187 shotgun metagenomic samples assembled into ~23.3 terabase pairs of contigs, a subset of which were binned into 1,158,468 medium- or high-quality MAGs, organized into 107,078 species-level clusters that partly overlap with those in proGenomes3.

Genomes from all three datasets were downloaded and taxonomically (re-)classified against GTDB r220 using GTDB-Tk version 2.4.0 (ref. [63]), and consensus taxonomies for species-level clusters were inferred based on adjusted majority votes, as described previously[23]. Compared with SPIRE version 1.0, 6,959 metagenomic samples were excluded for the present analysis based on data type and provenance (for example, excluding samples explicitly enriched for viruses) or insufficient assembly size. For the remaining 92,187 samples, habitat annotations and contextual data were updated by manually curating against an extended microntology version 0.3.0 encompassing 103 terms and categories (ref. [64]; reference set and updated annotations available via https://spire.embl.de/downloads and Supplementary Table 2) and further organized into 32 higher-level categories (Supplementary Table 1).

## Estimates of Bacterial and Archaeal diversity based on taxonomic marker genes

The majority of analyses presented in the main text are based on 168 near-universal taxonomic marker genes, as established by the GTDB: 120 Bacterial (bac120; ref. 13) and 53 Archaeal (arc53; ref. 14) markers, with an overlap of five genes. We downloaded profile hidden Markov models for these marker sets as part of the GTDB-Tk version 2.4.0 database and used the HMMER version 3.4 (ref. 65) hmmsearch routine to identify and extract marker gene sequences among predicted open reading frames. For the final filtered set of marker genes (see below), this yielded on average $904,654 \pm 98,879$ sequences per Bacterial marker in proGenomes3, $93,198 \pm 4,733$ among GTDB r220 species representatives, $840,023 \pm 133,904$ in medium- and high-quality SPIRE MAGs and $3,051,465 \pm 1,290,024$ among unbinned SPIRE contigs (that is, those not assigned to a MAG passing genome quality filters in SPIRE version 1). For Archaea, we obtained $2,451 \pm 150$ sequences from proGenomes3, $4,898 \pm 410$ from GTDB r220 species representatives, $15,504 \pm 2,593$ from SPIRE MAGs and $107,419 \pm 75,823$ from unbinned SPIRE contigs (Supplementary Table 3 provides a per-gene overview).

For each marker gene, we clustered extracted sequences using the MMseqs2 cascading clustering routine[66] with a cutoff of 96.5% sequence similarity (see below) and otherwise default parameters. To convert the number of marker gene sequence clusters into a corresponding number of species-level clusters, we estimated conversion factors as follows. First, we generated marker gene cluster discovery curves via iterative logarithmic rarefaction (that is, we downsampled the number of considered gene sequences along a logarithmic scale (10, 20, …, 100 sequences; 200, 300, …, 1,000 sequences and so on) with ten iterations at each step). At each rarefaction point, we recorded the number of discovered marker gene sequence clusters and the number of represented species or species-level genome clusters in proGenomes3, GTDB r220 and among SPIRE MAGs, considering each data source individually. We then used linear regression models of the type number_of_species - number_of_gene_clusters with a forced intercept at 0 along these rarefactions to estimate gene-cluster-to-species conversion factors (that is, the number of newly discovered species per newly discovered marker gene cluster). Extended Data Fig. 8 shows the resulting fits for ten randomly selected marker genes, each for Archaea and Bacteria. Based on benchmarks of marker gene sequence similarity cutoffs ranging from 95.0 to 99.5%, we based further analyses on 96.5% clusters as these showed very robust linear fits (with standard errors in the range of $10^{-3}$) with conversion factors closest to identity (that is, roughly one species discovered per marker gene cluster discovered), and high consistency across the different species-level reference clusterings in the underlying datasets (based on 40 SpecI marker genes in proGenomes3, 95% whole-genome average nucleotide identity in GTDB and a combination of both approaches among SPIRE MAGs). Finally, we filtered the set of considered marker genes by excluding redundant markers (part of both the arc53 and bac120 sets and therefore prone to cross-mappings) and removing markers for which predicted species numbers deviated by more than 20% from the domain-level median (in particular for some Archaeal markers, detected sequence counts were inflated, possibly due to the detection of Bacterial paralogues). Sequence counts per data source for the remaining set of 28 Archaeal and 102 Bacterial marker genes are available in Supplementary Table 3.

## Inference of global and habitat-resolved species discovery curves

For each of 32 broadly defined habitat categories (ranging from 267–19,659 samples per group; Supplementary Table 1), as well as for the combined set of all 92,187 metagenomes in SPIRE version 1.1, we generated Bacterial and Archaeal species discovery (or rarefaction) curves. We iteratively subsampled the number of considered metagenomes along a logarithmic rarefaction scale, with five random permutations

per step. At each rarefaction point and for each considered marker gene, we inferred the number of discovered species based on the number of discovered marker gene clusters, using gene-specific conversion factors, as described above. Rarefaction permutations were averaged per marker gene and then summarized per marker gene set (bac120 for Bacteria and arc53 for Archaea) as median predicted species counts at each rarefaction step across marker genes (Fig. 1). Extended Data Fig. 1 shows relative errors (coefficient of variation; that is, the standard deviation divided by the mean) at each rarefaction step for each marker gene; consistently, the coefficient of variation decreased to <5% when including 1,000–10,000 samples for Bacteria (that is, when using ~1–10% of the total data), with drops at slightly higher rarefaction steps for Archaea.

In an independent approach, we performed an incremental rarefaction, stratified by habitats: we sequentially added samples from each of the 32 broadly defined habitat categories, starting with adult, healthy human gut samples, then randomized the discovery order within each habitat block and tracked the globally discovered species (Extended Data Fig. 5 and Supplementary Figs. 1 and 2). In other words, we tracked the contribution of each habitat to overall discovered species diversity, beyond the diversity discovered in previously considered habitats.

Counts of discovered marker gene clusters (and, by proxy, inferred discovered species) were hierarchically stratified by data source to account for overlap between datasets as follows: all clusters containing at least one sequence originating from a genome in proGenomes3 were labelled as proGenomes3, irrespective of the origin of other sequences within the same cluster (data series marked in dark blue in main Fig. 1 and Extended Data Fig. 5); clusters containing sequences from GTDB r220 (and any other data source except proGenomes3) were labelled as GTDB (light blue in Fig. 1 and Extended Data Fig. 5); clusters containing sequences from SPIRE MAGs (but neither from proGenomes3 nor GTDB) were labelled as SPIRE MAG (orange data series); clusters containing only sequences from unbinned SPIRE contigs were labelled as unbinned, non-singleton if they contained at least two sequences from different contigs (dark yellow data series) and unbinned, all otherwise (light yellow data series). Thus, clusters labelled as GTDB represent sequence diversity contained in GTDB r220 beyond that already represented in proGenomes3, and a substantial subset of proGenomes3-labelled clusters also contained sequences originating from GTDB, SPIRE MAGs or unbinned contigs. Similarly, SPIRE MAG clusters corresponded to diversity not already covered by proGenomes3 and GTDB r220, and unbinned clusters encompassed diversity not represented in any genome or MAG in the dataset. Moreover, by design we only tracked marker gene clusters in metagenomic samples, meaning that a non-negligible subset of genome-based clusters from proGenomes3 and GTDB r220 were not represented in rarefaction runs and calculations, as corresponding sequences were not detected in any of the 92,187 considered metagenomic assemblies (main text).

## Calculation of species discovery coefficients

We quantified the rate at which species-level clusters are discovered in each habitat (and globally, across all habitats) using the equation

$$S = k \times N^{-\gamma} \tag{1}$$

where $S$ is the number of newly discovered species per $N$ samples added to the survey; $k$ is a proportionality constant; and $\gamma$ is a saturation coefficient. An analogous formula is commonly used to describe a (microbial) pangenome's openness[67,68] (that is, the degree to which genes are discovered as more genomes from the same species are considered). Moreover, the approach is conceptually and mathematically related to the Heridan–Heaps law in linguistics, which describes the number of distinct words in a document as a function of that document's length[69].

We fit equation (1) to each habitat-specific and global rarefaction curve described above, stratified by data source, resulting in estimates for $\gamma$ for each considered marker gene; Extended Data Fig. 9 shows the example fits for a randomly selected subset of marker genes. For more intuitive interpretability, we calculated a species discovery coefficient $\alpha$ as

$$\alpha = 1 - \gamma \qquad (2)$$

and summarized values across marker genes within each domain-specific set as median values. Thus defined, $\alpha$ scales on $[-\infty, 1]$, although only mildly negative values are expected to be observed in practice. If $\alpha \leq 0$, species discovery in a habitat is saturated, meaning that adding more samples of the same type is not expected to add species to the survey, analogous to a closed microbial pangenome where additional strains do not add genes. Values for $\alpha$ in $[0, 1]$ correspond to unsaturated species discovery curves where additional samples continue to add species to the survey (analogous to open pangenomes). Lower $\alpha$ values indicate a more pronounced flattening off in the rarefaction curve, indicating a more pronounced slowdown in species discovery, and higher $\alpha$ values indicate a less pronounced decrease in the rate of species discovery. For $\alpha \to 1$, species discovery is fully unsaturated, meaning that each newly added sample adds species to the survey, with no discernible flattening off in the species discovery curve.

### Phylogenetic inference and analyses

Phylogenetic trees for each marker gene were constructed as follows. We first realigned cluster representative amino acid sequences against the respective marker gene hidden Markov models using the hmmalign routine from HMMER version 3.4 (ref. [65]), then we trimmed the resulting pseudo-multiple sequence alignments to informative columns using clipkit version 1.4.1 (ref. [70]) in kpi-gappy mode and removed sequences with >70% gaps among the remaining columns. Based on the resulting alignments, we inferred phylogenetic trees using FastTree 2 version 2.1.11 (ref. [71]) under the Whelan–Goldman model[72].

We calculated RED values[32] for each node in the resulting trees using castor package version 1.8.3 in R[73]. To infer marker-gene-specific RED cutoffs at the phylum, class, order, family and genus levels, we subset phylogenies to sequences originating from fully taxonomically classified genomes in proGenomes3, GTDB r220 or among SPIRE MAGs, cut the trees at incremental RED values (with a RED tolerance of ±0.1) and compared the resulting partitions (that is, tip sets) versus assigned taxonomic labels, quantifying partition consistency as adjusted mutual information[74]. These steps were performed iteratively on trees re-rooted using each recognized phylum in turn as an outgroup and then summarized across alternative root positions (ignoring the current outgroup) to obtain average RED values per internal node, in a workflow adapted from ref. [32].

After confirming that the resulting RED cutoffs at peak adjusted mutual information (that is, best capturing the distribution of established taxonomic labels) well approximated those reported in ref. [32] across individual Archaeal marker genes, we used GTDB's reference cutoffs (with a ±0.1 tolerance interval) for Bacterial marker gene trees and cut the full trees at the respective levels. The number of resulting clusters (that is, nodes where the tree was cut) and their composition (that is, each node's descendant tips) were then used as estimates of clade-level groups at each taxonomic level, for each marker gene. We manually selected genes that showed the most consistent and robust profiles across iterations (and rootings) and whose phylogenies were not ostensibly impacted by paralogues, bringing the final sets of considered genes to 29 Archaeal and 53 Bacterial markers. Marker gene phylogenies were visualized using the ggtree and ggtreeExtra packages in R[75].

Marker gene trees were also used for the analyses underlying cumulative marker gene distributions, with the results presented in Extended Data Fig. 4. Internal nodes corresponding to phyla recognized in the GTDB were identified as the most recent common ancestors of tips representing marker gene clusters containing genes from reference genomes or MAGs classified to the focal phylum. Phylum-level subtrees cut at the respective most recent common ancestor nodes were then profiled for the ratio of tips representing only unbinned genes to tips representing genomes (that is, binned genes), as well as for average genomic features (GC content, genome size and coding density) and the estimated total number of represented species. To alleviate the effects of genome misclassification and errors in phylogenetic inference, analyses were repeated five times for each phylum, each time using a random subset of 80% of genomes to identify the most recent common ancestor node.

### Fitting taxonomic clade size distributions

Four datasets were used to fit clade sizes to previously suggested distributions: (1) the above-defined clade-level groups (based on RED cutoffs), for each Archaeal and Bacterial marker gene; (2) GTDB r226 Archaeal and Bacterial reference taxonomies (accessed via https://gtdb.ecogenomic.org/); (3) the GBIF backbone taxonomy for eukaryotes, Bacteria, Archaea and viruses[76], accessed using the taxizedb R package[77]; and (4) the Microbe Atlas Project[11] 16S rRNA reference database version 3.0 (ref. [41]), using 97% OTUs to represent species and 90% OTUs to represent genus. For each of these taxonomies, we first fit Willis curves[29] relating the frequency of clades to their size (that is, the number of subclades they contained) as a regular power law:

$$C = a \times S^{-\omega} \qquad (3)$$

where $C$ is the number of higher clades (for example, genus) containing $S$ subclades (for example, species); $a$ is a proportionality constant; and $\omega$ is the scaling coefficient, referred to as the Willis coefficient here for simplicity. In other words, equation (3) relates the frequency of a clade to its size, with positive $\omega$ values indicating that large clades (for example, genera containing many species) are exponentially less common than small clades (for example, genera containing few or only one species). Otherwise, given the similarity of equation (3) to equation (1) above, the interpretation of $\omega$ is closely related to that of the above-defined species discovery coefficient $\alpha$.

We moreover fit clade sizes to the distribution first suggested by Yule[30] specifically for this purpose and later formalized by Simon[78]:

$$f(C_S) = \rho \times B(C_S, \rho + 1) \qquad (4)$$

where $f(C_S)$ is the non-negative frequency of clades of size $S$ (that is, containing $S$ subclades); B is the beta function; and $\rho > 0$ is the Yule–Simon shape parameter. The distribution results from a Yule process where newly arising subclades preferentially attach to larger existing clades, proportionally to the existing clades' sizes, sometimes colloquially referred to as a rich-get-richer process. For sufficiently large counts $C_S$, the frequency $f(C_S)$ approximates a power law as in equation (3), with coefficient $\omega$ at $\sim\rho + 1$. Parameter $\rho$ indicates the shape of the distribution and the strength of preferential attachment, with a large $\rho$ value indicating a sharper drop in the distribution (less pronounced preferential attachment, a less tail-heavy distribution and higher dominance by small clades), whereas a smaller $\rho$ value indicates more tail-heavy distributions and stronger preferential attachment effects (large clades are larger; small clades are fewer). In other words, $\rho$ can be thought of as a rich-get-richer coefficient, with low $\rho$ values indicating a stronger dominance of fewer large clades in the clade size distribution.

Habitat-stratified analyses (Fig. 4c) were conducted by recomputing clade size distributions using only genes assembled from samples of the focal habitat. For phylum-resolved analyses (Fig. 4d), subtrees corresponding to recognized phyla were iteratively extracted from

full marker gene phylogenies (see above) and clade size distributions were recomputed. In both cases, estimated $\rho$ values were summarized as the median across marker genes.

## Reporting summary

Further information on research design is available in the Nature Portfolio Reporting Summary linked to this article.

## Data availability

Metagenomic assemblies, MAGs, gene calls and corresponding annotations, and marker gene sequences for the ar53 and bac122 sets extracted from SPIRE assemblies, proGenomes3 and GTDB r220 are available from http://spire.embl.de/downloads. Pre-processed and derived data are available via Zenodo (https://zenodo.org/records/17482698; ref. 79). Inferred marker gene phylogenies with annotations, as well as pre-generated tree visualizations for Archaeal markers are available via the European Bioinformatics Institute BioStudies repository under accessions S-BSST2111 (ref. 80), S-BSST2112 (ref. 81), S-BSST2113 (ref. 82), S-BSST2116 (ref. 83) and S-BSST2117 (ref. 84).

## Code availability

Supporting analysis code has been deposited at https://github.com/grp-schmidt/ms-census.

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

## Acknowledgements

We thank A. N. Orakov (Harvard T. H. Chan School of Public Health) and J. Walter (University College Cork) for helpful discussions and feedback on the manuscript. This research was conducted with financial support from Research Ireland (grant 12/RC/2273-P2 to V.P.P.K. and T.S.B.S.) and the Australian Research Council (grant FT230100724 to L.P.C.). A.S. has received funding from the European Research Council under the European Union's Horizon 2020 research and innovation programme (grant agreement 947317; ASymbEL), which also supported the position of O.M.M. This work made use of EMBL IT Services high-performance computing resources (https://zenodo.org/records/12785830)[85]. The authors are deeply grateful to P.B. for his guidance and contributions to this work and beyond.

## Author contributions

O.M.M., P.B. and T.S.B.S. conceived of the study idea. V.P.P.K., O.M.M., A.F. and T.S.B.S. conducted the initial computational analyses. V.P.P.K., O.M.M., A.F., D.R.M., E.K., L.P.C., A.S. and T.S.B.S. analysed the data. A.S. and P.B. provided additional resources for computational analyses. V.P.P.K. and T.S.B.S. designed the display items. T.S.B.S. wrote the initial manuscript draft. All authors reviewed, edited and revised the manuscript.

## Funding

## Competing interests

The authors declare no competing interests.

## Additional information

**Extended data** is available for this paper at https://doi.org/10.1038/s41564-026-02314-6.

**Correspondence and requests for materials** should be addressed to Thomas S. B. Schmidt.

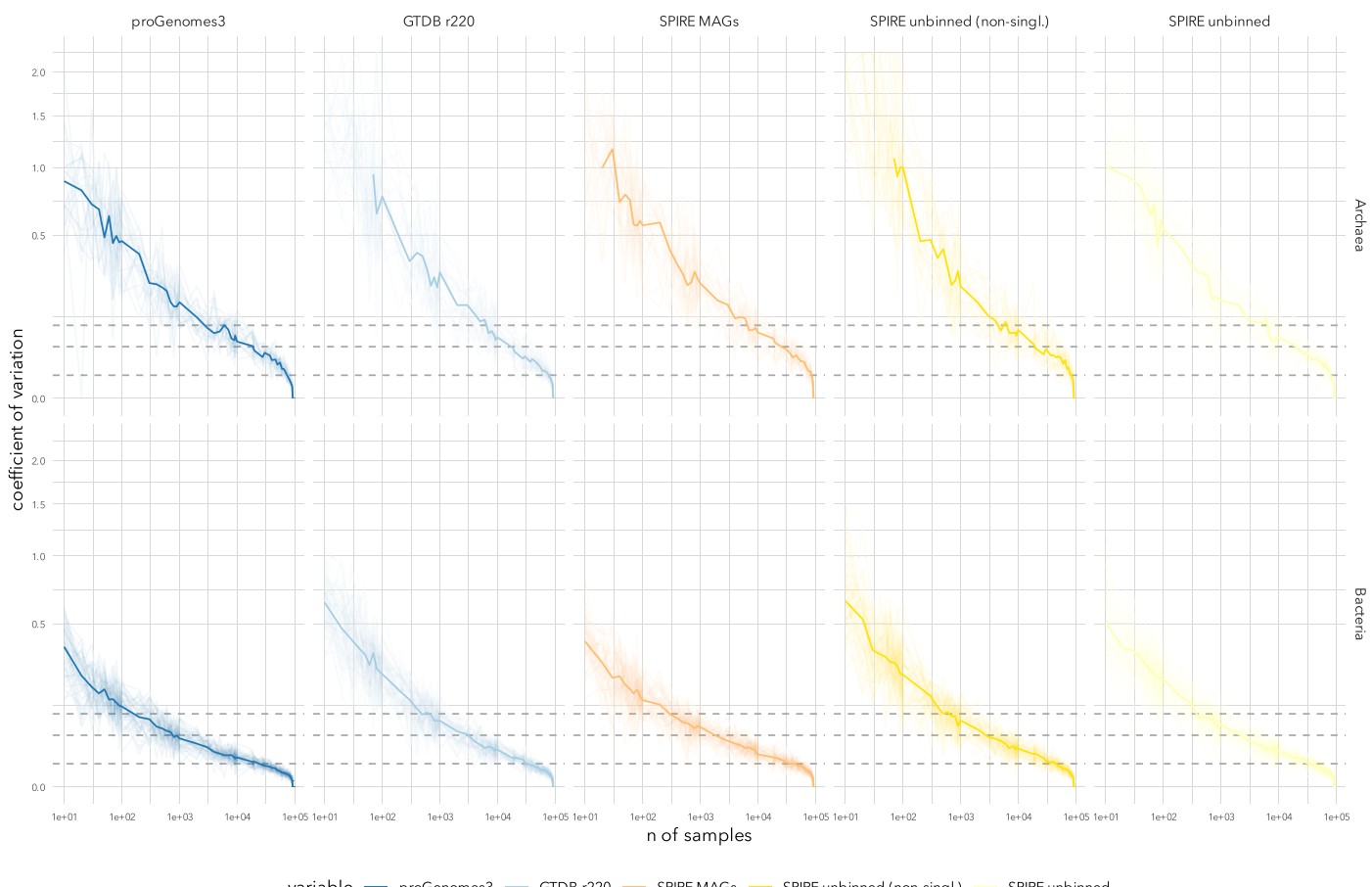

**Extended Data Fig. 1 | Uncertainty and noise in marker gene-based rarefaction curves.** The coefficient of variation (*cv*; standard deviation divided by mean; y axis) across 5 random permutations is shown per rarefaction step (x axis) for the different tested data sources. Thin data series correspond to individual marker genes, the median across marker genes is emphasized. Dotted lines indicate reference levels of 1%, 5% and 10% cv. Note that the sharp drop in cv towards the right is a mathematical necessity as noise between permutations decreases once nearly all samples are included.

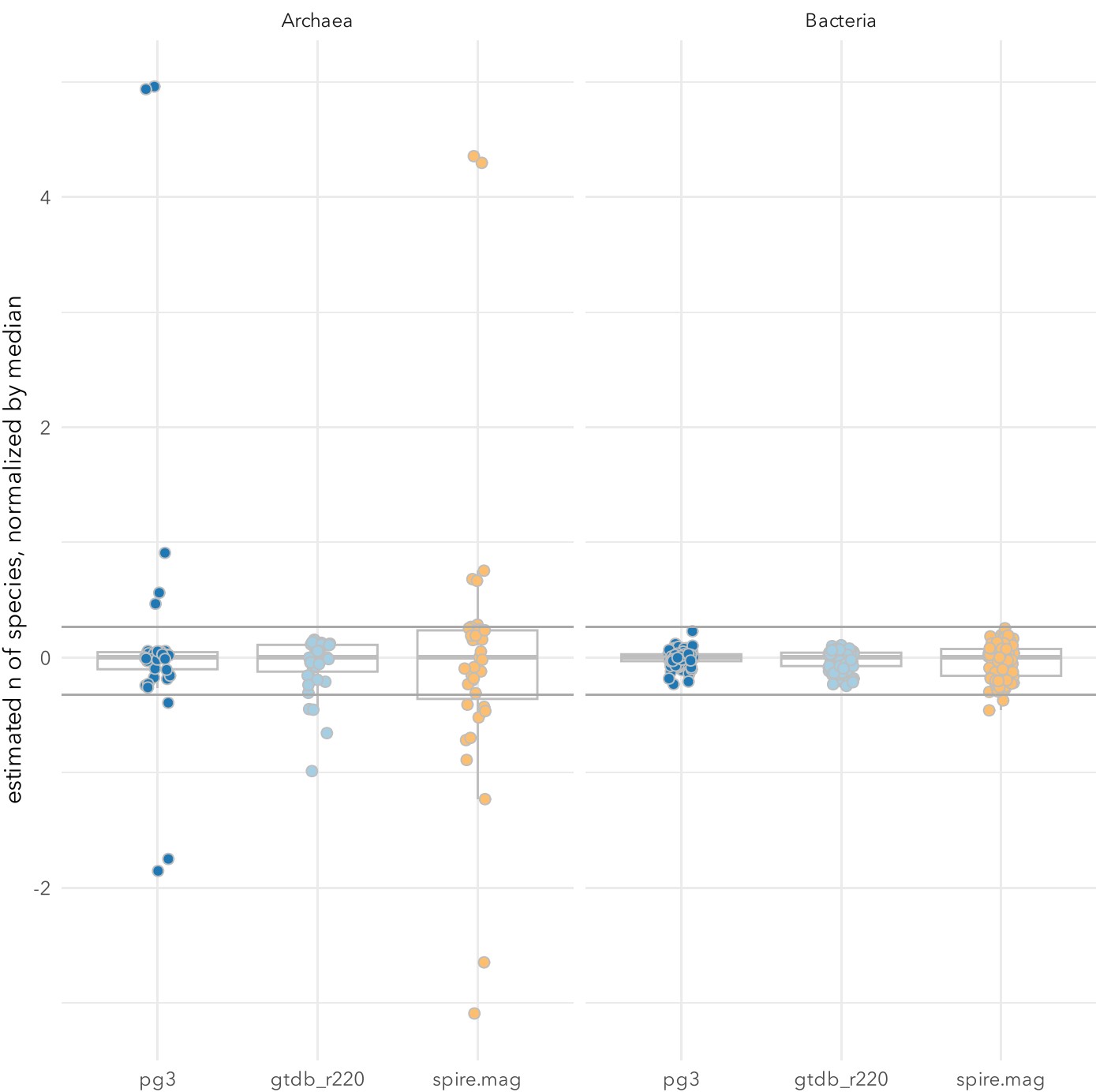

**Extended Data Fig. 2 | Species count estimates are remarkably consistent across different marker genes.** Each dot represents the estimated number of species based on observed marker gene clusters (n = 53 for Archaea, n = 122 for Bacteria; see Methods) for one marker gene, normalized by the median across marker genes within categories. Archaeal outliers to the top were marker genes with putative cross-mapping to Bacterial orthologs (leading to an overestimation of diversity). For analyses in the main text, only a subset of consistent marker genes, further filtered based on additional criteria (see Methods) were used. Boxplots indicate median values, 25th and 75th percentiles (boxes) and 1,5x interquartile ranges (whiskers).

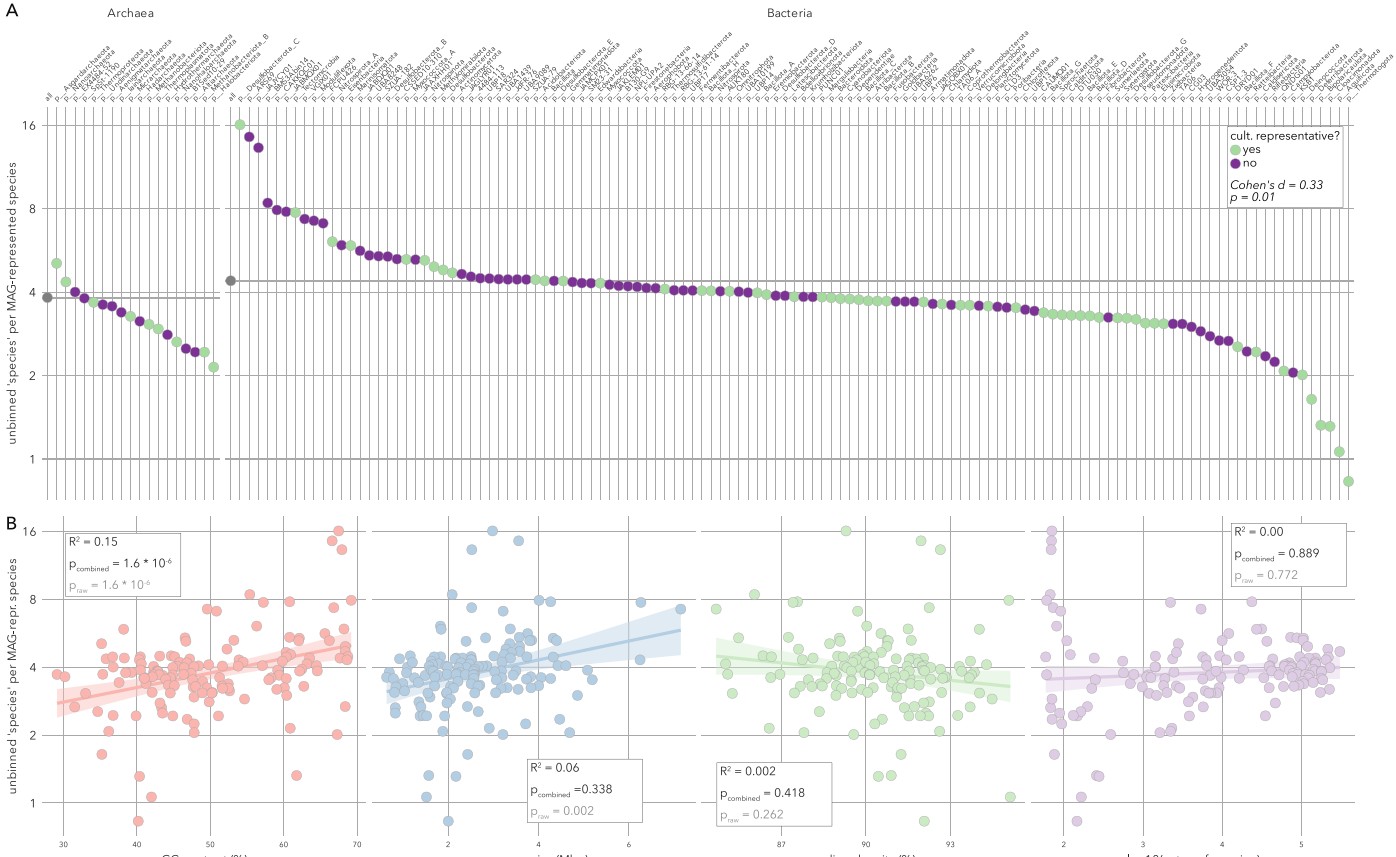

**Extended Data Fig. 3 | Unbinned genes are enriched among uncultivated phyla and associated with genomic GC content. (a)** The ratio of tips (that is, marker gene clusters) containing only unbinned genes to those containing genes from reference genomes or MAGs (y axis) is shown for subtrees of individual recognized phyla, sampled iteratively from full marker gene phylogenies (see Methods). This unbinned ratio was higher for phyla without (purple) compared to those with (green) cultured representatives (Cohen's d = 0.33; two-sided Wilcoxon p = 0.01). **(b)** Ratio of unbinned to binned markers per phylum (y axis) shown against average genomic GC content, genome size, coding density (all derived from GTDB reference genomes per phylum) and the estimated number of species per phylum. $R^2$ and p values (two-sided t test) are shown for individual linear regressions of unbinned ratios against each variable, as well as for a combined multiple regression in which only GC content stood out as significantly associated. Shaded bands indicate 95% confidence intervals of linear fits.

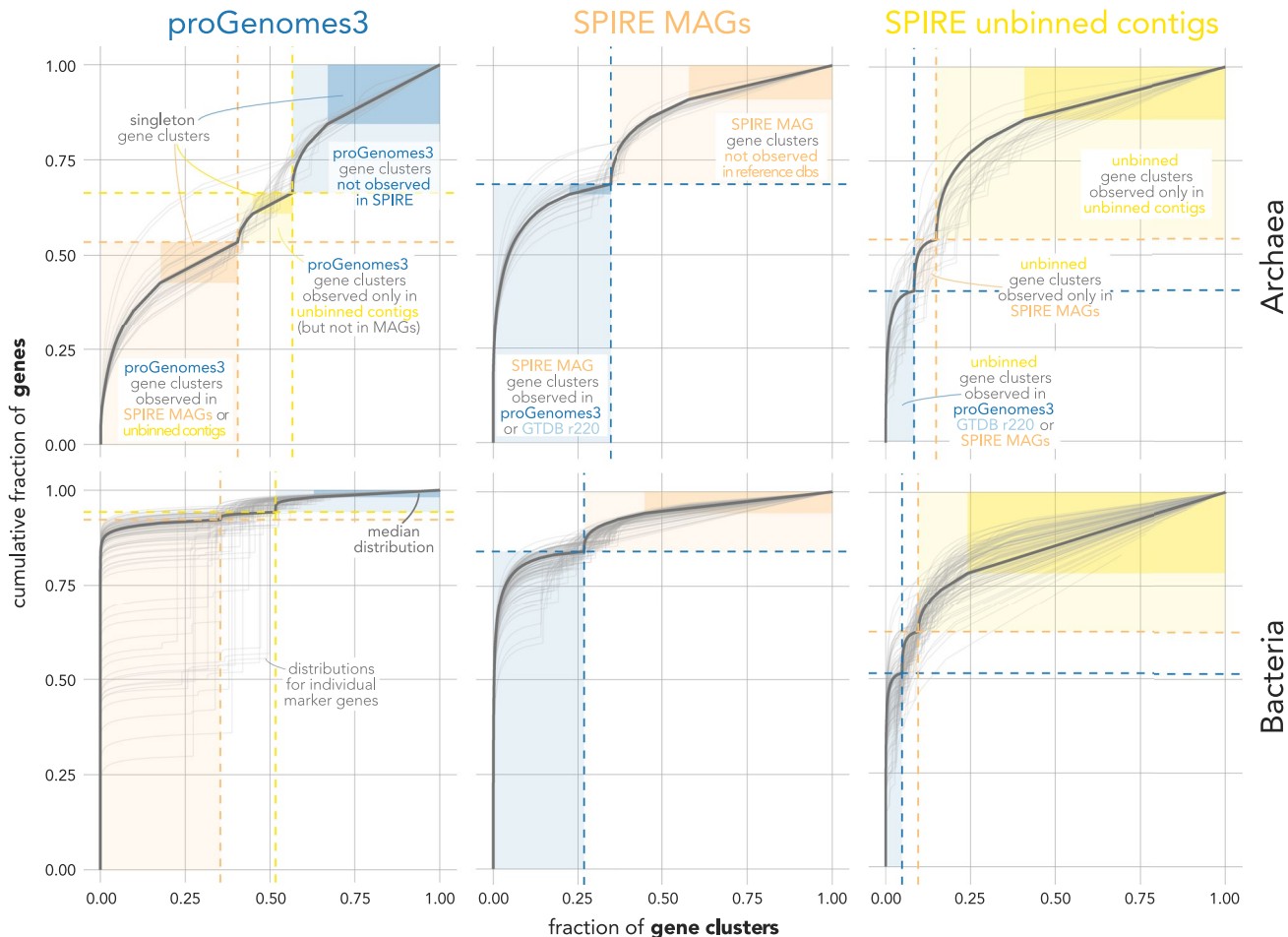

**Extended Data Fig. 4 | Cumulative distributions of marker genes per marker gene cluster.** The cumulative fraction of marker genes (y axis) contained in ranked marker gene clusters (x axis) is shown for individual marker genes (grey lines) and the median marker gene (emphasis). Emphasized rectangles indicate the part of the distribution corresponding to singleton marker gene clusters. The left panels show distributions for marker genes extracted from proGenomes3 reference genomes: the leftmost part (orange) corresponds to gene clusters also observed in SPIRE MAGs or unbinned contigs; the middle section (yellow) correspond to clusters observed in unbinned contigs, but not MAGs; the rightmost section (blue) are clusters not observed in metagenomic assemblies. The central panels show distributions for marker genes extracted from SPIRE MAGs: marker gene clusters that were also observed in proGenomes3 or GTDB r220 genomes (blue, left) or not (right, orange). The rightmost panels show distributions for marker genes extracted from unbinned contigs: gene clusters also observed in reference genome sets (blue, left); those observed in SPIRE MAGs, but not reference genome sets (yellow, central); and those observed exclusively in unbinned contigs.

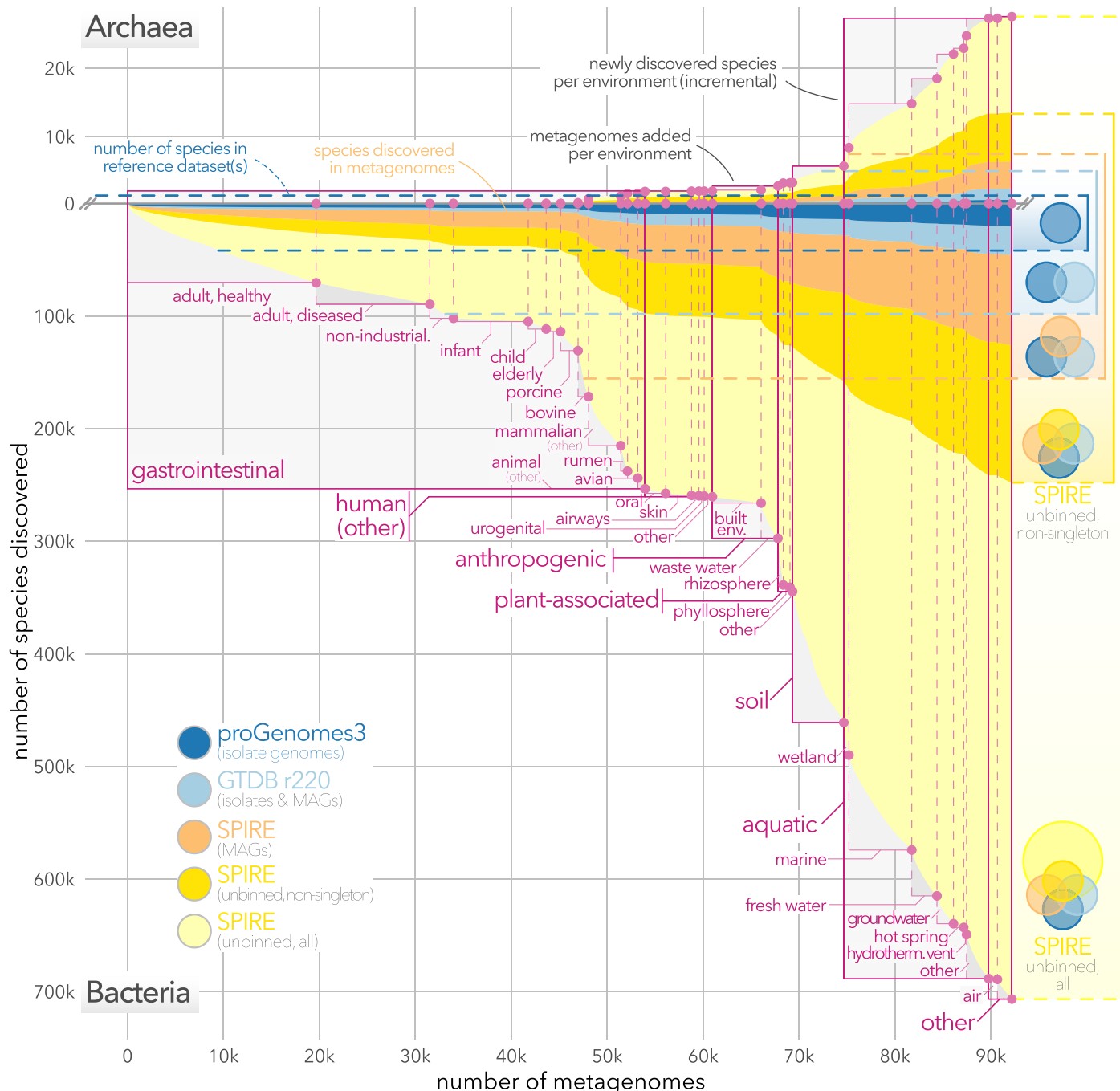

**Extended Data Fig. 5 | Soils and aquatic habitats are major reservoirs of unexplored microbial diversity.** 'Habitat-cumulative' rarefaction curves for Archaea (top) and Bacteria (bottom). Samples were added to the survey sequentially by habitat and newly discovered species were tracked *incrementally*, that is in addition to those contained in previously added habitats (see Methods). Horizontal dashed lines indicate reference levels, that is species counts including those not detectable in our metagenomic dataset.

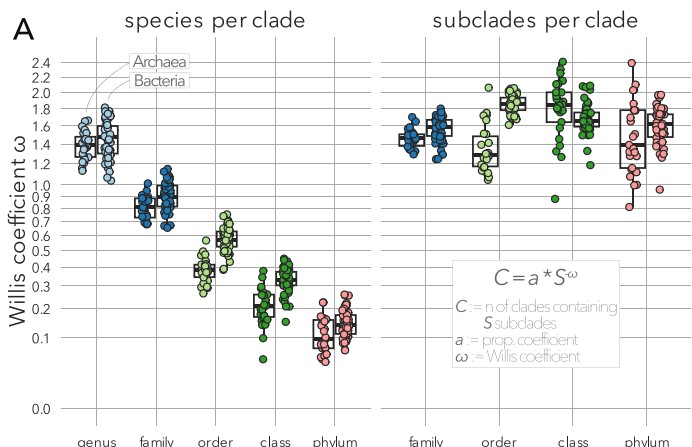

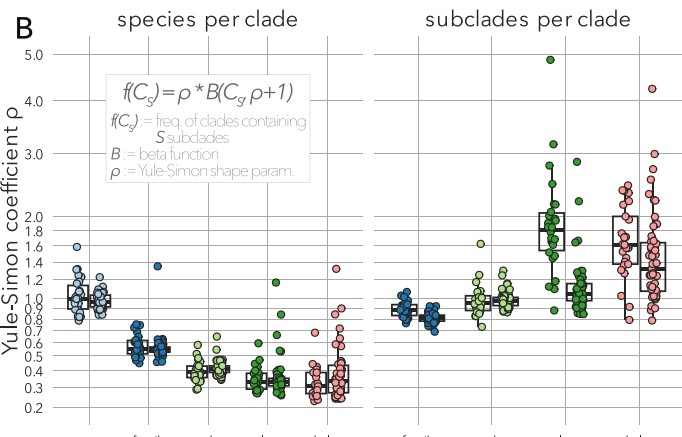

**Extended Data Fig. 6 | Willis and Yule-Simon coefficients across marker genes and taxonomic levels.** Estimated Willis coefficients ω (**a**) and Yule-Simon coefficients ρ (**b**) across taxonomic level. Each dot represents an estimate for one individual Archaeal (n = 29, left series) or Bacterial (n = 53, right series) marker gene. Data is provided for both 'species per clade' (clade size distributions defined as the total number of species per clade, that is per phylum, class, etc) and 'subclades per clade' (that is, genera per family, families per order, etc). Boxplots indicate median values, 25th and 75th percentiles (boxes) and 1,5x interquartile ranges (whiskers).

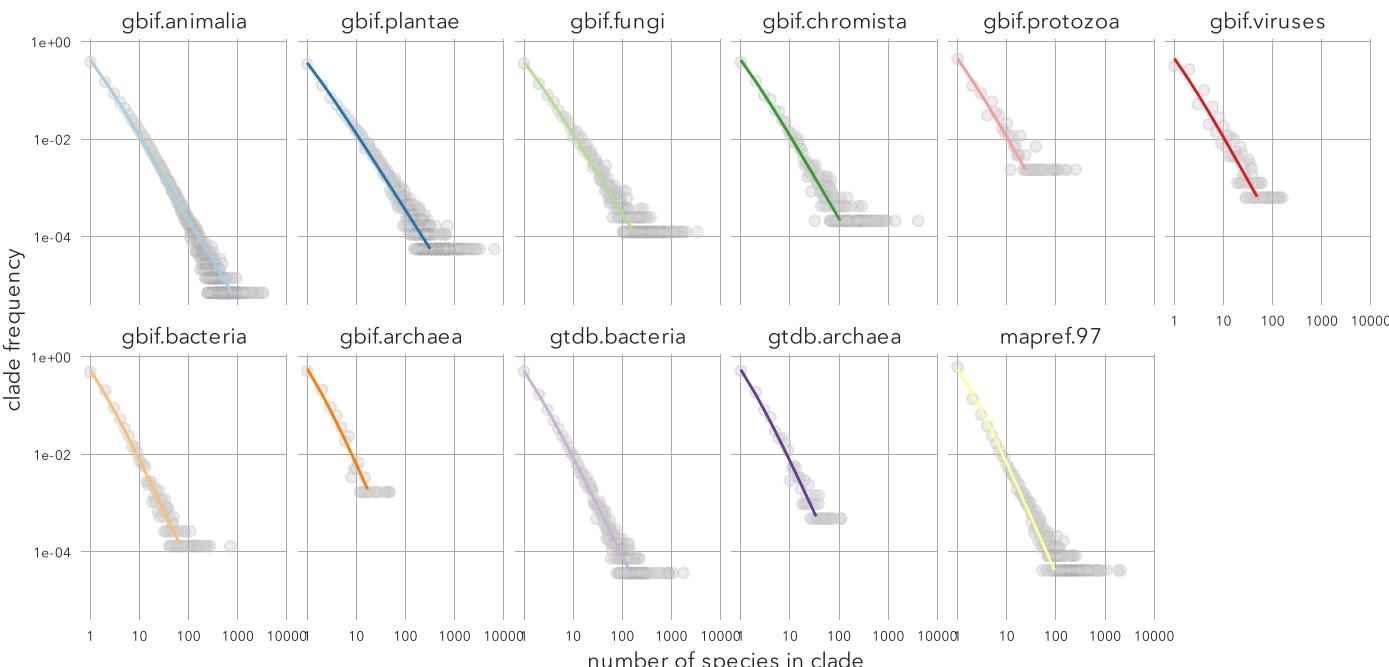

**Extended Data Fig. 7 | Yule curves based on external reference taxonomies.**
Log-log plots of clade sizes (number of species per clade; x axis) against clade counts (number of clades containing x species; y axis), analogous to Fig. 4a. Lines correspond to fitted Yule curves. Each data series corresponds to a reference taxonomy from the Global Biodiversity Information Facility (GBIF), the GTDB r226 or the Microbe Atlas Project (see Methods).

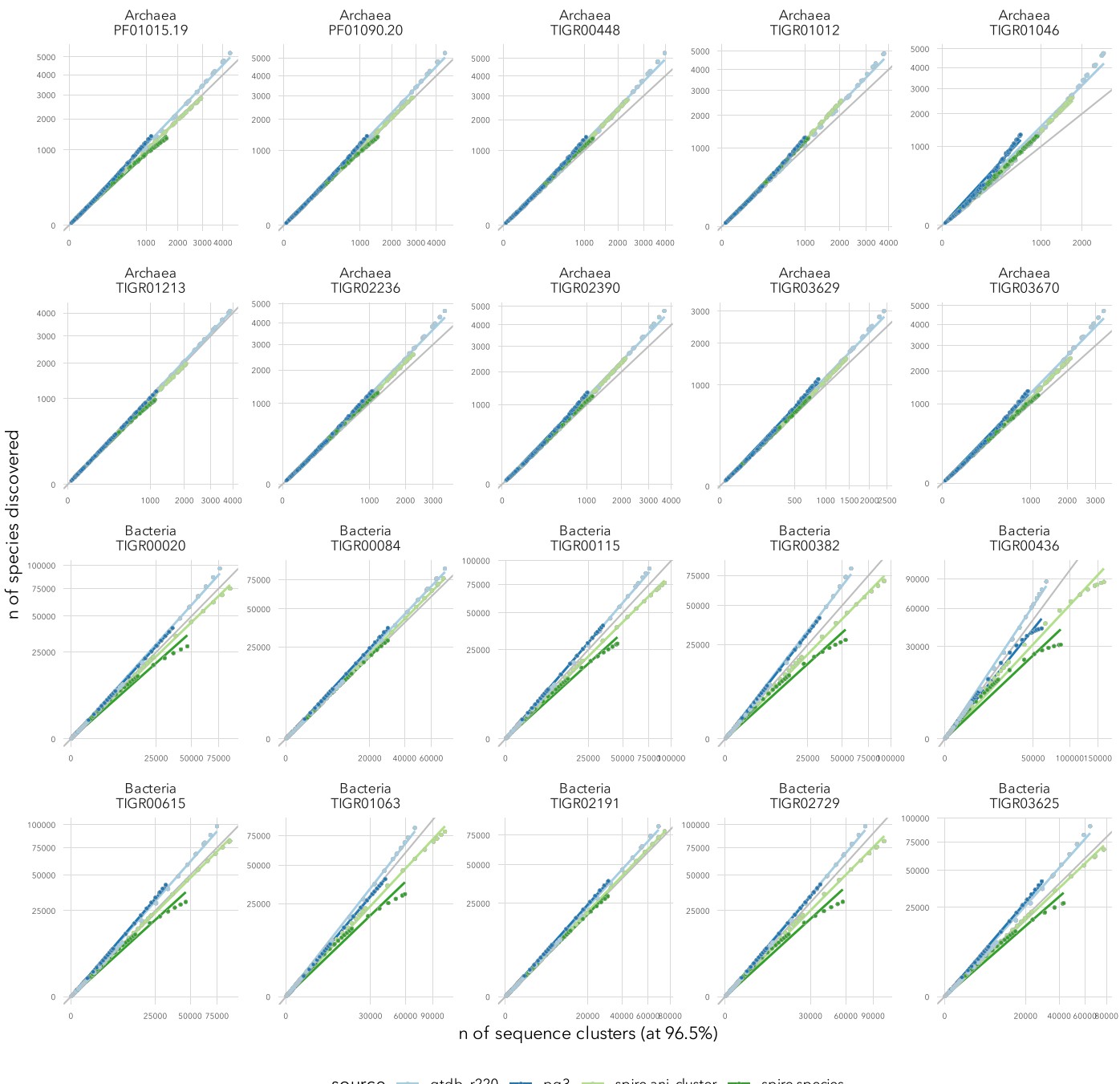

**Extended Data Fig. 8 | Marker gene-specific parameters to estimate species counts.** Plots show the number of discovered species (y axis) against the number of sampled marker gene sequence clusters (x axis), for 10 randomly picked marker genes each for Archaea and Bacteria. The underlying data was obtained from iterative rarefaction along logarithmic steps (see Methods) for four different data series: 'GTDB r220' refers to (clustered) marker genes originating from GTDB r220 reference genomes, and corresponding species definitions; 'pg3' are marker genes extracted from proGenomes3, and corresponding GTDB-based species classifications; 'spire.ani_cluster' uses SPIRE MAGs clustered at 95% average nucleotide identity as 'species' clusters (that is, analogous to GTDB definitions) and MAG-derived marker genes; 'spire.species' uses species-level GTDB classifications of SPIRE MAGs instead. The deviation of the 'spire.species' series at higher marker gene cluster counts is likely due to 'underclassification' (a substantial fraction of SPIRE MAGs were not classifiable to species level against the GTDB reference). Therefore, in analyses shown in the main text, coefficients fitted from the 'spire.ani_cluster' series were used to estimate species counts from (unbinned) marker gene clusters.

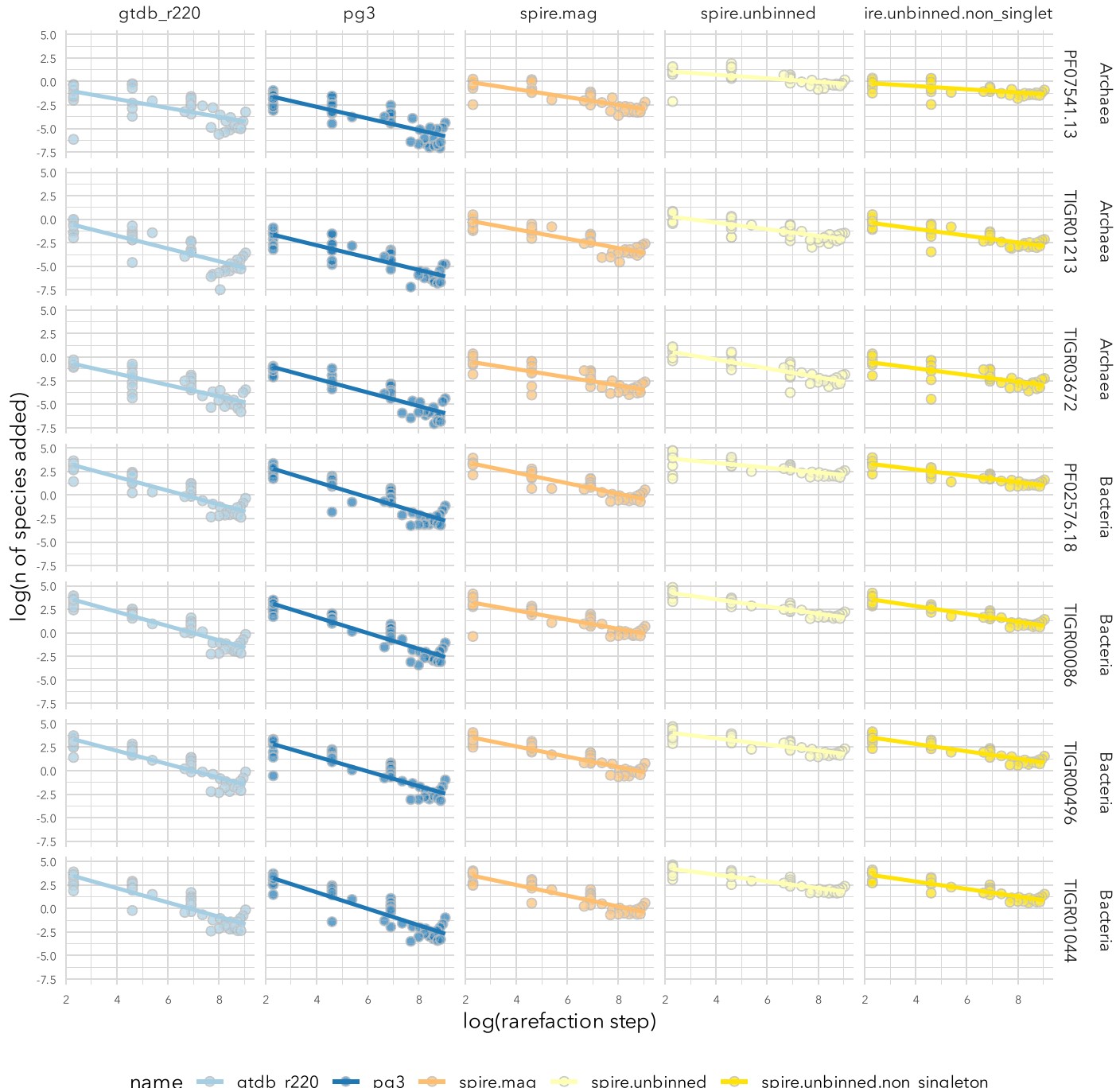

**Extended Data Fig. 9 | Fitting species discovery coefficients.** Plots show the underlying data to fit 'species discovery coefficients' α (see Methods) for seven randomly picked marker genes for Archaea and Bacteria. Based on rarefactions, the log-log plots show the number of added species per sample (that is, number of additionally 'discovered' species, y axis) per rarefaction step (that is, number of samples added to the survey, x axis). Species discovery coefficients were estimated from the corresponding regressions in log-log space.

# Reporting Summary

## Statistics

For all statistical analyses, confirm that the following items are present in the figure legend, table legend, main text, or Methods section.

| n/a | Confirmed | |
|---|---|---|
| ☐ | ☒ | The exact sample size (*n*) for each experimental group/condition, given as a discrete number and unit of measurement |
| ☒ | ☐ | A statement on whether measurements were taken from distinct samples or whether the same sample was measured repeatedly |
| ☐ | ☒ | The statistical test(s) used AND whether they are one- or two-sided<br>*Only common tests should be described solely by name; describe more complex techniques in the Methods section.* |
| ☒ | ☐ | A description of all covariates tested |
| ☐ | ☒ | A description of any assumptions or corrections, such as tests of normality and adjustment for multiple comparisons |
| ☐ | ☒ | A full description of the statistical parameters including central tendency (e.g. means) or other basic estimates (e.g. regression coefficient) AND variation (e.g. standard deviation) or associated estimates of uncertainty (e.g. confidence intervals) |
| ☐ | ☒ | For null hypothesis testing, the test statistic (e.g. *F*, *t*, *r*) with confidence intervals, effect sizes, degrees of freedom and *P* value noted<br>*Give P values as exact values whenever suitable.* |
| ☒ | ☐ | For Bayesian analysis, information on the choice of priors and Markov chain Monte Carlo settings |
| ☒ | ☐ | For hierarchical and complex designs, identification of the appropriate level for tests and full reporting of outcomes |
| ☐ | ☒ | Estimates of effect sizes (e.g. Cohen's *d*, Pearson's *r*), indicating how they were calculated |

*Our web collection on statistics for biologists contains articles on many of the points above.*

## Software and code

Policy information about availability of computer code

| Data collection | SPIRE v1<br>proGenomes v3<br>GTDB r220<br>GTDB r226<br>HMMer v3.4<br>GTDB-tk v2.4.0<br>microntology v0.3<br>mapref v3.0<br>GBIF taxonomy via taxizedb package, acc. 07-2025 |
|---|---|
| Data analysis | Commented analysis code was uploaded to a GitHub repository: https://github.com/grp-schmidt/ms-census<br>All external tools used for analyses are detailed, with version numbers, in the main text.<br>ttax-<br>MMSeqs2 v15.6f452<br>HMMER v3.4<br>clipkit v1.4.1<br>FastTree2 v2.1.11<br>castor v1.8.3<br>taxizedb v0.3.2 |

For manuscripts utilizing custom algorithms or software that are central to the research but not yet described in published literature, software must be made available to editors and reviewers. We strongly encourage code deposition in a community repository (e.g. GitHub). See the Nature Portfolio guidelines for submitting code & software for further information.

# Data

Policy information about <u>availability of data</u>

All manuscripts must include a <u>data availability statement</u>. This statement should provide the following information, where applicable:

- Accession codes, unique identifiers, or web links for publicly available datasets
- A description of any restrictions on data availability
- For clinical datasets or third party data, please ensure that the statement adheres to our <u>policy</u>

Source data was derived from the SPIRE database (spire.embl.de) where it was prepared as described in https://doi.org/10.1093/nar/gkad943. External datasets were obtained from the proGenomes (v3) and GTDB (r220) databases, as detailed in the Methods section of the main text. Genes for analyses were extracted via HMMs (using HMMer v3) as described in the main text. No custom code was required for data collection; data curation (annotation of habitat information etc based on sample metadata) was perfomed as described in the primary SPIRE db reference and further detailed in the main text.
All data sources and tools are detailed, with version numbers and/or access dates, in the main text.

Metagenomic assemblies, Metagenome-Assembled Genomes (MAGs), gene calls and corresponding annotations are available via spire.embl.de/downloads. Extracted marker gene sequences for the ar53 and bac122 sets from SPIRE assemblies, proGenomes3 and GTDB r220 are likewise available via spire.embl.de/downloads. Pre-processed and derived data is available via Zenodo (https://zenodo.org/records/17482698). Inferred marker gene phylogenies with annotations, as well as pre-generated tree visualizations for archaeal markers are available via the EBI BioStudies repository under accessions S-BSST2111, S-BSST2112, S-BSST2113, S-BSST2116, and S-BSST2117.

# Research involving human participants, their data, or biological material

Policy information about studies with <u>human participants or human data</u>. See also policy information about <u>sex, gender (identity/presentation), and sexual orientation</u> and <u>race, ethnicity and racism</u>.

| | |
|---|---|
| Reporting on sex and gender | n.a. |
| Reporting on race, ethnicity, or other socially relevant groupings | n.a. |
| Population characteristics | n.a. |
| Recruitment | n.a. |
| Ethics oversight | n.a. |

Note that full information on the approval of the study protocol must also be provided in the manuscript.

# Field-specific reporting

Please select the one below that is the best fit for your research. If you are not sure, read the appropriate sections before making your selection.

☐ Life sciences  ☐ Behavioural & social sciences  ☒ Ecological, evolutionary & environmental sciences

For a reference copy of the document with all sections, see <u>nature.com/documents/nr-reporting-summary-flat.pdf</u>

# Ecological, evolutionary & environmental sciences study design

All studies must disclose on these points even when the disclosure is negative.

| | |
|---|---|
| Study description | A computational re-appraisal of discoverable diversity in metagenomically assembled contigs. Clustering >500M taxonomic marker gene sequences to species level, we track how many species-level groups in publicly available data are missed by common genome-centric approaches (including reference, isolate-based genomes and metagenome-assembled genomes). Moreover, based on large marker gene phylogenies, we provide estimates of how many additional deeper clades (genus to phylum) are 'hiding' in public data, but missed by current genome binning methods. Our analyses are stratified by microbial habitat and in particular provide habitat-specific 'discovery coefficients' that quantify the (differential) expectation of how many more lineages will be discovered as more metagenomic sequence data is added to the survey. Finally, we explored whether microbial clade size distributions follow empirical power laws, in line with century-old hypotheses on biodiversity. |
| Research sample | 23.2 Tbp of assembled metagenomic contigs, sourced from the SPIRE db (spire.embl.de), from which a subset of 92k well-annotated metagenomic samples (curated public data) were selected. In addition, we analysed the reference genome databases proGenomes3 and GTDB r220 that provide (curated and processed) datasets of publicly available prokaryotic genomes. From these datasets, we extracted 502M sequences for 130 established taxonomic marker genes on which all further analyses were based. |
| Sampling strategy | No sample size calculations were performed. Our survey of 92k metagenomes is (to our knowledge) the largest such dataset currently available. |

| | |
|---|---|
| Data collection | Data was obtained from publicly available sources as described above under 'Research sample'. |
| Timing and spatial scale | N/A |
| Data exclusions | We excluded several taxonomic marker genes that are either known to hit orthologs across multiple domains (e.g., models that hit both archaeal and bacterial genes) or showed problematic phylogenies with signs of (erroneously included) paralogs upon manual inspection. |
| Reproducibility | Analyses were re-run (with modified parameters) several times and iteratively, starting from the marker gene sequence clustering step. Analysis code is available via a GitHub repository (see above); intermediary datasets / analyses are available via dedicated repositories and via spire.embl.de/downloads. |
| Randomization | N/A |
| Blinding | N/A |

Did the study involve field work? ☐ Yes ☒ No

# Reporting for specific materials, systems and methods

We require information from authors about some types of materials, experimental systems and methods used in many studies. Here, indicate whether each material, system or method listed is relevant to your study. If you are not sure if a list item applies to your research, read the appropriate section before selecting a response.

## Materials & experimental systems

| n/a | Involved in the study |
|---|---|
| ☒ ☐ | Antibodies |
| ☒ ☐ | Eukaryotic cell lines |
| ☒ ☐ | Palaeontology and archaeology |
| ☒ ☐ | Animals and other organisms |
| ☒ ☐ | Clinical data |
| ☒ ☐ | Dual use research of concern |
| ☒ ☐ | Plants |

## Methods

| n/a | Involved in the study |
|---|---|
| ☒ ☐ | ChIP-seq |
| ☒ ☐ | Flow cytometry |
| ☒ ☐ | MRI-based neuroimaging |

## Plants

| | |
|---|---|
| Seed stocks | *Report on the source of all seed stocks or other plant material used. If applicable, state the seed stock centre and catalogue number. If plant specimens were collected from the field, describe the collection location, date and sampling procedures.* |
| Novel plant genotypes | *Describe the methods by which all novel plant genotypes were produced. This includes those generated by transgenic approaches, gene editing, chemical/radiation-based mutagenesis and hybridization. For transgenic lines, describe the transformation method, the number of independent lines analyzed and the generation upon which experiments were performed. For gene-edited lines, describe the editor used, the endogenous sequence targeted for editing, the targeting guide RNA sequence (if applicable) and how the editor was applied.* |
| Authentication | *Describe any authentication procedures for each seed stock used or novel genotype generated. Describe any experiments used to assess the effect of a mutation and, where applicable, how potential secondary effects (e.g. second site T-DNA insertions, mosiacism, off-target gene editing) were examined.* |

