## [Peer Review File · Nature Microbiology]

Unbinned contigs expand known diversity in the global microbiome

Corresponding Author: Dr Thomas Schmidt

Version 0:

Reviewer comments:

Reviewer #1

(Remarks to the Author)

Prasoodanan and colleagues present an analysis of microbial diversity from single copy marker genes on unbinned metagenomic contigs. They deem this diversity "discoverable", as these are taxa that have been sequenced, but not yet genomically characterized (i.e., binned into genomes from metagenomes). They estimate the degree of discoverable microbial diversity across different environments and at different taxonomic ranks. The paper is well-written and the figures are elegant.

However, I feel like the paper brings up some major questions that are left unanswered, and could be expanded to become more than a census. The limitation of the work as simply a census of discoverable microbial diversity is that the extent of discoverable microbial diversity is constantly in flux, as more samples are sequenced and better software tools for assembly and binning are developed and employed. Therefore, a census is simply a snapshot and will quickly become dated. However, the work lends itself to coming very close to answering some generalizable and important questions:

1. The primary question I am left with reading this work is: how much of the divergence between discoverable taxa and genomically characterized taxa is due to random chance vs systematic issues in current approaches in metagenomic assembly and binning? Are MAGs simply a random subsample of all unbinned sequences, and as we sequence and assemble more they will 'catch-up'? Or, are there some taxa that we are consistently undersampling in MAGs or missing entirely, possibly for biological reasons? For example: large genome sizes, high microdiversity, very repetitive genomes, even DNA modifications, may play a role here.

If the degree to which the divergence between discoverable and genomically characterized taxa is systematic can be quantified, that would be a very valuable contribution to the field.

The authors should just caution that this divergence will be methodologically dependent, e.g., long reads, single cells, and better algorithms might capture taxa that are currently systematically missed.

2. At line 84, the authors suggest that the discoverable yet hidden taxa in this data could be part of the "rare biosphere". But, I think they are missing an analysis of this: are the discoverable taxa consistently rare? How does rarity, both within sample and between samples, influence a taxon's ability to have been genomically characterized or not? Are we missing just the rare things, or are there some taxa that are ubiquitous yet rarely abundant, or are there some that are even abundant but still being missed? An analysis of this would greatly increase the impact of the work.

3. The authors report a fascinating novel deep archaeal lineage that is also very abundant (they have many species in this lineage). Are they able to use the data to tell us more about this lineage? Where is it found? How abundant is it in these samples? For example, it seems like it is sometimes found in the human gut (is that true in Figure 3?)?! Are they able to identify samples which perhaps co-assembly of would result in a genome? This would demonstrate how characterizing discoverable taxa can help guide their eventual genomic characterization.

4. The authors compute rarefaction curves but they do not compute their error. Are they able to do bootstrapping to put some error bars on rarefaction curves (maybe for supplementary)? Can they then estimate the number of entirely undiscovered taxa (e.g., extrapolating out the curves, perhaps using an approach like breakaway) using both unbinned and binned taxa?

5. I really would take issue with the title "beyond genome-centric approaches". I really think the authors should change this title, because they have not shown that these discoverable taxa are in fact "beyond" it, merely that they haven't yet been characterized. Eventually, these taxa will be characterized by genome-centric approaches, whether it is with Illumina, long-read, single cell, cultivation, etc. "Beyond" seems to incorrectly imply forever out of reach. I would instead suggest "yet unidentified by

genome-centric approaches" or something similar.

6. Figure 1: I think it'd make more sense for the SPIRE (MAGS) category to be blue, not orange. Because this is a "genome" category, more like proGenomes3 and GTDB r220.

7. Please add the code to a GitHub repository associated with the work for easier reproduction.

(Remarks on code availability)

I cannot reproduce this analysis with code in PDF format, but upon visual inspection it seems reasonable.

Reviewer #2

(Remarks to the Author)

In this manuscript, Prasoodanan and Maistrenko et al. use the previously developed SPIRE platform to assess how much prokaryotic novelty remains to be discovered in existing sequence databases. They conduct gene-level analyses of metagenomic assemblies, which enables detection of organisms that cannot be recovered via traditional binning approaches. The major conclusions are that many prokaryotic genomes remain to be discovered in existing datasets, and that environmental sequencing yields more novelty than additional sequencing of the gastrointestinal tract.

This manuscript addresses an interesting and important question in microbiology, and the analysis appears to be exceptionally robust. However, the manuscript is built around a single central question (albeit a worthwhile one) and some aspects lacked clarity. With some revisions, I believe this work would be of sufficiently broad interest to warrant publication in Nature Microbiology.

Sincerely,
Matt Olm, CU Boulder

Specific comments:

1) The in-line figures in the main PDF sent to reviewers are low resolution and difficult to read; Figure 4 is entirely unreadable. I was eventually able to view the figures using the separate figure PDF files, but the core equations did not render anywhere (e.g., lines 507 and 520), and I could not find a way to view them.

2) The central question of the manuscript is: How much additional prokaryotic diversity can be detected in metagenomic assemblies by examining genes rather than relying on genome bins? This is an interesting and well-addressed question. However, I believe the manuscript would benefit from a more concise format- perhaps a short-format article with two figures. Figures 1, 2, and 4 all convey the same main point (that most novelty is found outside the human gastrointestinal tract), and consolidating them could improve impact and clarity.

3) While I'm convinced the core methods are sound, I believe additional supplemental figures would help readers understand the approach. For example, visualizations of the "very robust linear fits" mentioned on line 458, the species discovery curves on line 471, and the fitted lines referenced on line 516 would all be useful. And of course, the equations on lines 507 and 520 need to be made readable.

4) An investigation into why these genes aren't binned would strengthen the study. Are they low coverage? Do they fall into low-quality bins, or no bins? Do they exhibit higher microdiversity than binned genes? Have you tried manually examining a few representative genes to understand why they failed to bin? Also, it would be helpful to include data on the frequency of species detected by novel genes beyond simply counting singletons and non-singletons. Finally, you might consider exploring why certain frequently assembled-but-never-binned genes consistently evade binning.

5) A comparison to singleM or sandpiper would be valuable. A quantitative comparison would be ideal (especially if you analyzed overlapping datasets), but even a brief discussion would suffice.

6) Why was RNA polymerase E chosen for Figure 3? It seems like a somewhat arbitrary choice.

7) I appreciated the inclusion of the caveats described on lines 257-262 and shown in Table 1. This is an important and well-executed control.

8) Please consider adding species information to Table 1. This would help readers interpret Figures 1 and 2.

9) As experts in the field, do you believe that the singleton genes represent genuine biological sequences or assembly artifacts? My instinct is that many are likely artifacts given the known limitations of assemblers and how distinct the singleton line appears. I'm not questioning the accuracy of the manuscript's text, but I got the impression that the authors may believe most singleton sequences are real. This comes through in statements like "non-singletons represent a lower bound," whereas I might describe them as a best estimate since the same misassembly could feasibly occur twice.

Suggestion (non-essential):

Several figures could benefit from streamlining. In Figure 1A and related figures, I would suggest emphasizing your source descriptions on line 329 (cultivated organisms (proGenomes3), reference MAGs (GTDB r220) or dataset-specific MAGs

(SPIRE)), rather than emphasizing the database names. Additionally, I recommend removing the “Unbinned all” category, since it’s less emphasized in the text and stretches the y-axis, making other differences harder to see. Finally, consider removing gridlines from several plots to make the visuals feel cleaner and less cluttered.

(Remarks on code availability)

Kudos to the authors for making their code available- this is an important step toward transparency and reproducibility. However, the code is provided as a single text large text file of R code with no README, usage instructions, or list of dependencies. While sharing the code in any form is far better than not sharing it at all, the current format would make it quite difficult for others to reproduce the analysis or build upon it (which may be acceptable, depending on the intended audience).

Reviewer #3

(Remarks to the Author)

General comments

This study has considerable value: not so much in the essentially preliminary analysis of undiscovered taxa, but in highlighting the issue and in providing access to a toolset that can be used by other researchers in a much more focussed manner.

Minor comments

Line 60. I disagree that ‘surveys of prokaryote diversity increasingly rely on isolate and MAGS’. Literature surveys will show that, for better or worse, 16S surveys still dominate, but are increasingly supplemented with MAG phylogenetics (reflecting the burgeoning of newly available technologies). I would argue that isolate-dependent surveys continue to grow, but at a fairly slow linear rate.

Line 194 and on: The authors might make the point, when discussing the relatively low taxon discovery rate in human associated metagenome data that this is perhaps not surprising, since humans, despite their diversity and complexity, represent a single species. The authors make a good point about the comparative heterogeneity of other habitats (e.g., see lines 371-373).

Line 267: The authors might consider explaining how their estimates of novel phylum level clades have any validity, considering the very high standard deviations.

(Remarks on code availability)

The Code

Comments from Dr Pedro Lebre (Senior Postdoctoral Fellow, University of Pretoria: pedro.lebre@up.ac.za)

I checked the code and the syntax seems perfectly fine – it’s harder to assess how it actually works without having the raw datasets, as from the scripts it looks like the authors used a bunch of pre-loaded Rstudio objects, which don't really have any correlation with the data available to the reader.

They might need to clarify in more detail where these initial objects originated from.

Decision Letter:

18th August 2025

Dear Dr Schmidt,

Thank you for your patience while your manuscript "A census of hidden and discoverable microbial diversity beyond genome-centric approaches" was under peer-review at Nature Microbiology. It has now been seen by 3 referees, whose expertise and comments you will find at the end of this email. Although they find your work of some potential interest, they have raised a number of concerns that will need to be addressed before we can consider publication of the work in Nature Microbiology.

In particular, the reviewers raise a number of important points related to ensuring the robustness of your results with additional analysis. Furthermore, while the reviewers do suggest shortening the paper, we do NOT feel that the work would fulfill our editorial criteria for a Brief Communication, so we would not encourage you to shorten the manuscript to that format, but rather to hone in on the more interesting discoveries and expound upon those. We would be open to considering this paper as a Resource, rather than an Article. Finally, the reviewers also raised a number of concerns about the code and its usability--this will need to be addressed in full before we can return the paper to reviewers (code should be fully accessible, documented with a README and able to be run with test datasets by the reviewers).

Should further experimental data allow you to address these criticisms, we would be happy to look at a revised manuscript.

We strongly support public availability of data. Please place the data used in your paper into a public data repository, if one exists, or alternatively, present the data as Source Data or Supplementary Information. If data can only be shared on request, please explain why in your Data Availability Statement, and also in the correspondence with your editor. For some data types, deposition in a public repository is mandatory - more information on our data deposition policies and available repositories can

be found at <https://www.nature.com/nature-research/editorial-policies/reporting-standards#availability-of-data>.

Please include a data availability statement as a separate section after Methods but before references, under the heading "Data Availability". This section should inform readers about the availability of the data used to support the conclusions of your study. This information includes accession codes to public repositories (data banks for protein, DNA or RNA sequences, microarray, proteomics data etc...), references to source data published alongside the paper, unique identifiers such as URLs to data repository entries, or data set DOIs, and any other statement about data availability. At a minimum, you should include the following statement: "The data that support the findings of this study are available from the corresponding author upon request", mentioning any restrictions on availability. If DOIs are provided, we also strongly encourage including these in the Reference list (authors, title, publisher (repository name), identifier, year). For more guidance on how to write this section please see: <http://www.nature.com/authors/policies/data/data-availability-statements-data-citations.pdf>

* If you have not done so already we suggest that you begin to revise your manuscript so that it conforms to our Article format instructions at <http://www.nature.com/nmicrobiol/info/final-submission>. Refer also to any guidelines provided in this letter.

When submitting the revised version of your manuscript, please pay close attention to our [href="https://www.nature.com/nature-portfolio/editorial-policies/image-integrity">Digital Image Integrity Guidelines.](https://www.nature.com/nature-portfolio/editorial-policies/image-integrity) and to the following points below:

EXTENDED DATA FIGURES

Link Redacted

Note: This url links to your confidential homepage and associated information about manuscripts you may have submitted or be reviewing for us. If you wish to forward this e-mail to co-authors, please delete this link to your homepage first.

Nature Microbiology is committed to improving transparency in authorship. As part of our efforts in this direction, we are now requesting that all authors identified as 'corresponding author' on published papers create and link their Open Researcher and Contributor Identifier (ORCID) with their account on the Manuscript Tracking System (MTS), prior to acceptance. This applies to primary research papers only. ORCID helps the scientific community achieve unambiguous attribution of all scholarly contributions. You can create and link your ORCID from the home page of the MTS by clicking on 'Modify my Springer Nature account'. For more information please visit [please visit www.springernature.com/orcid](http://www.springernature.com/orcid).

If you wish to submit a suitably revised manuscript we would hope to receive it within 6 months. If you cannot send it within this time, please let us know. We will be happy to consider your revision, even if a similar study has been accepted for publication at Nature Microbiology or published elsewhere (up to a maximum of 6 months).

Yours sincerely,

Reviewer Expertise:

Referee #1: metagenomics, bioinformatics, microbiome
Referee #2: microbiome, bioinformatics, microbial ecology
Referee #3: environmental microbial ecology, biodiversity

Reviewer Comments:

Reviewer #1 (Remarks to the Author):

Prasoodanan and colleagues present an analysis of microbial diversity from single copy marker genes on unbinned metagenomic contigs. They deem this diversity "discoverable", as these are taxa that have been sequenced, but not yet genomically characterized (i.e., binned into genomes from metagenomes). They estimate the degree of discoverable microbial diversity across different environments and at different taxonomic ranks. The paper is well-written and the figures are elegant.

However, I feel like the paper brings up some major questions that are left unanswered, and could be expanded to become more than a census. The limitation of the work as simply a census of discoverable microbial diversity is that the extent of discoverable microbial diversity is constantly in flux, as more samples are sequenced and better software tools for assembly and binning are developed and employed. Therefore, a census is simply a snapshot and will quickly become dated. However, the work lends itself to coming very close to answering some generalizable and important questions:

1. The primary question I am left with reading this work is: how much of the divergence between discoverable taxa and genomically characterized taxa is due to random chance vs systematic issues in current approaches in metagenomic assembly and binning? Are MAGs simply a random subsample of all unbinned sequences, and as we sequence and assemble more they will 'catch-up'? Or, are there some taxa that we are consistently undersampling in MAGs or missing entirely, possibly for biological reasons? For example: large genome sizes, high microdiversity, very repetitive genomes, even DNA modifications, may play a role here.

If the degree to which the divergence between discoverable and genomically characterized taxa is systematic can be quantified, that would be a very valuable contribution to the field.

The authors should just caution that this divergence will be methodologically dependent, e.g., long reads, single cells, and better algorithms might capture taxa that are currently systematically missed.

2. At line 84, the authors suggest that the discoverable yet hidden taxa in this data could be part of the "rare biosphere". But, I think they are missing an analysis of this: are the discoverable taxa consistently rare? How does rarity, both within sample and between samples, influence a taxon's ability to have been genomically characterized or not? Are we missing just the rare things, or are there some taxa that are ubiquitous yet rarely abundant, or are there some that are even abundant but still being missed? An analysis of this would greatly increase the impact of the work.

3. The authors report a fascinating novel deep archaeal lineage that is also very abundant (they have many species in this lineage). Are they able to use the data to tell us more about this lineage? Where is it found? How abundant is it in these samples? For example, it seems like it is sometimes found in the human gut (is that true in Figure 3?)?! Are they able to identify samples which perhaps co-assembly of would result in a genome? This would demonstrate how characterizing discoverable taxa can help guide their eventual genomic characterization.

4. The authors compute rarefaction curves but they do not compute their error. Are they able to do bootstrapping to put some error bars on rarefaction curves (maybe for supplementary)? Can they then estimate the number of entirely undiscovered taxa (e.g., extrapolating out the curves, perhaps using an approach like breakaway) using both unbinned and binned taxa?

5. I really would take issue with the title "beyond genome-centric approaches". I really think the authors should change this title, because they have not shown that these discoverable taxa are in fact "beyond" it, merely that they haven't yet been characterized. Eventually, these taxa will be characterized by genome-centric approaches, whether it is with Illumina, long-read, single cell, cultivation, etc. "Beyond" seems to incorrectly imply forever out of reach. I would instead suggest "yet unidentified by genome-centric approaches" or something similar.

6. Figure 1: I think it'd make more sense for the SPIRE (MAGS) category to be blue, not orange. Because this is a "genome" category, more like proGenomes3 and GTDB r220.

7. Please add the code to a GitHub repository associated with the work for easier reproduction.

Reviewer #1 (Remarks on code availability):

I cannot reproduce this analysis with code in PDF format, but upon visual inspection it seems reasonable.

Reviewer #2 (Remarks to the Author):

In this manuscript, Prasoodanan and Maistrenko et al. use the previously developed SPIRE platform to assess how much

prokaryotic novelty remains to be discovered in existing sequence databases. They conduct gene-level analyses of metagenomic assemblies, which enables detection of organisms that cannot be recovered via traditional binning approaches. The major conclusions are that many prokaryotic genomes remain to be discovered in existing datasets, and that environmental sequencing yields more novelty than additional sequencing of the gastrointestinal tract.

This manuscript addresses an interesting and important question in microbiology, and the analysis appears to be exceptionally robust. However, the manuscript is built around a single central question (albeit a worthwhile one) and some aspects lacked clarity. With some revisions, I believe this work would be of sufficiently broad interest to warrant publication in *Nature Microbiology*.

Sincerely,
Matt Olm, CU Boulder

Specific comments:

1) The in-line figures in the main PDF sent to reviewers are low resolution and difficult to read; Figure 4 is entirely unreadable. I was eventually able to view the figures using the separate figure PDF files, but the core equations did not render anywhere (e.g., lines 507 and 520), and I could not find a way to view them.

2) The central question of the manuscript is: How much additional prokaryotic diversity can be detected in metagenomic assemblies by examining genes rather than relying on genome bins? This is an interesting and well-addressed question. However, I believe the manuscript would benefit from a more concise format- perhaps a short-format article with two figures. Figures 1, 2, and 4 all convey the same main point (that most novelty is found outside the human gastrointestinal tract), and consolidating them could improve impact and clarity.

3) While I'm convinced the core methods are sound, I believe additional supplemental figures would help readers understand the approach. For example, visualizations of the "very robust linear fits" mentioned on line 458, the species discovery curves on line 471, and the fitted lines referenced on line 516 would all be useful. And of course, the equations on lines 507 and 520 need to be made readable.

4) An investigation into why these genes aren't binned would strengthen the study. Are they low coverage? Do they fall into low-quality bins, or no bins? Do they exhibit higher microdiversity than binned genes? Have you tried manually examining a few representative genes to understand why they failed to bin? Also, it would be helpful to include data on the frequency of species detected by novel genes beyond simply counting singletons and non-singletons. Finally, you might consider exploring why certain frequently assembled-but-never-binned genes consistently evade binning.

5) A comparison to singleM or sandpiper would be valuable. A quantitative comparison would be ideal (especially if you analyzed overlapping datasets), but even a brief discussion would suffice.

6) Why was RNA polymerase E chosen for Figure 3? It seems like a somewhat arbitrary choice.

7) I appreciated the inclusion of the caveats described on lines 257-262 and shown in Table 1. This is an important and well-executed control.

8) Please consider adding species information to Table 1. This would help readers interpret Figures 1 and 2.

9) As experts in the field, do you believe that the singleton genes represent genuine biological sequences or assembly artifacts? My instinct is that many are likely artifacts given the known limitations of assemblers and how distinct the singleton line appears. I'm not questioning the accuracy of the manuscript's text, but I got the impression that the authors may believe most singleton sequences are real. This comes through in statements like "non-singletons represent a lower bound," whereas I might describe them as a best estimate since the same misassembly could feasibly occur twice.

Suggestion (non-essential):

Several figures could benefit from streamlining. In Figure 1A and related figures, I would suggest emphasizing your source descriptions on line 329 (cultivated organisms (proGenomes3), reference MAGs (GTDB r220) or dataset-specific MAGs (SPIRE)), rather than emphasizing the database names. Additionally, I recommend removing the "Unbinned all" category, since it's less emphasized in the text and stretches the y-axis, making other differences harder to see. Finally, consider removing gridlines from several plots to make the visuals feel cleaner and less cluttered.

Reviewer #2 (Remarks on code availability):

Kudos to the authors for making their code available- this is an important step toward transparency and reproducibility. However, the code is provided as a single text large text file of R code with no README, usage instructions, or list of dependencies. While sharing the code in any form is far better than not sharing it at all, the current format would make it quite difficult for others to reproduce the analysis or build upon it (which may be acceptable, depending on the intended audience).

Reviewer #3 (Remarks to the Author):

General comments

This study has considerable value: not so much in the essentially preliminary analysis of undiscovered taxa, but in highlighting the issue and in providing access to a toolset that can be used by other researchers in a much more focussed manner.

Minor comments

Line 60. I disagree that ‘surveys of prokaryote diversity increasingly rely on isolate and MAGS’. Literature surveys will show that, for better or worse, 16S surveys still dominate, but are increasingly supplemented with MAG phylogenetics (reflecting the burgeoning of newly available technologies). I would argue that isolate-dependent surveys continue to grow, but at a fairly slow linear rate.

Line 194 and on: The authors might make the point, when discussing the relatively low taxon discovery rate in human associated metagenome data that this is perhaps not surprising, since humans, despite their diversity and complexity, represent a single species. The authors make a good point about the comparative heterogeneity of other habitats (e.g., see lines 371-373).

Line 267: The authors might consider explaining how their estimates of novel phylum level clades have any validity, considering the very high standard deviations.

Reviewer #3 (Remarks on code availability):

The Code

Comments from Dr Pedro Lebre (Senior Postdoctoral Fellow, University of Pretoria: pedro.lebre@up.ac.za)

I checked the code and the syntax seems perfectly fine – it’s harder to assess how it actually works without having the raw datasets, as from the scripts it looks like the authors used a bunch of pre-loaded Rstudio objects, which don't really have any correlation with the data available to the reader.

They might need to clarify in more detail where these initial objects originated from.

Version 1:

Reviewer comments:

Reviewer #1

(Remarks to the Author)

I commend the authors on their thoughtful responses and significant updates to their manuscript. They have answered all of my questions and added considerably to the manuscript.

(Remarks on code availability)

The new code on Github is accessible and much improved.

Reviewer #2

(Remarks to the Author)

The authors have adequately addressed all of my previous critiques and I now believe the manuscript is suitable for publication in Nature Microbiology.

(Remarks on code availability)

The code is very well organized and more than sufficient to fully understand and recapitulate the authors findings.

Reviewer #3

(Remarks to the Author)

None

(Remarks on code availability)

Decision Letter:

Our ref: NMICROBIOL-25062281A

20th January 2026

Dear Dr. Schmidt,

Thank you for submitting your revised manuscript "Hidden but discoverable diversity in the global microbiome." (NMICROBIOL-25062281A). It has now been seen by the original referees and their comments are below. The reviewers find that the paper has improved in revision, and therefore we'll be happy in principle to publish it in Nature Microbiology, pending minor revisions to satisfy the referees' final requests and to comply with our editorial and formatting guidelines.

Thank you again for your interest in Nature Microbiology Please do not hesitate to contact me if you have any questions.

Sincerely,

Reviewer #1 (Remarks to the Author):

I commend the authors on their thoughtful responses and significant updates to their manuscript. They have answered all of my questions and added considerably to the manuscript.

Reviewer #1 (Remarks on code availability):

The new code on Github is accessible and much improved.

Reviewer #2 (Remarks to the Author):

The authors have adequately addressed all of my previous critiques and I now believe the manuscript is suitable for publication in Nature Microbiology.

Reviewer #2 (Remarks on code availability):

The code is very well organized and more than sufficient to fully understand and recapitulate the authors findings.

Reviewer #3 (Remarks to the Author):

None

Version 2:

Decision Letter:

4th March 2026

Dear Dr Schmidt,

I am pleased to accept your Resource "Unbinned contigs expand known diversity in the global microbiome" for publication in Nature Microbiology. Thank you for having chosen to submit your work to us and many congratulations.

You may wish to make your media relations office aware of your accepted publication, in case they consider it appropriate to organize some internal or external publicity. Once your paper has been scheduled you will receive an email confirming the publication details. This is normally 3-4 working days in advance of publication. If you need additional notice of the date and time of publication, please let the production team know when you receive the proof of your article to ensure there is sufficient time to coordinate. Further information on our embargo policies can be found here:

<https://www.nature.com/authors/policies/embargo.html>

Authors may need to take specific actions to achieve compliance with funder and institutional open access mandates. If your research is supported by a funder that requires immediate open access (e.g. according to [Plan S principles](https://www.springernature.com/gp/open-science/plan-s-compliance) or the [NIH public access policy](https://www.springernature.com/gp/open-science/us-federal-agency-compliance)) then you should select the gold OA route, and we will direct you to the compliant route where possible. Because authors warrant under our subscription licensing terms that they haven't committed to licensing any version of their article under a licence inconsistent with the terms of our agreement – including the applicable embargo period – publication under the subscription model isn't suitable for authors whose funders require no embargo.

With kind regards,

P.S. Click on the following link if you would like to recommend Nature Microbiology to your librarian
<http://www.nature.com/subscriptions/recommend.html#forms>

** Visit the Springer Nature Editorial and Publishing website at http://editorial-jobs.springernature.com?utm_source=ejP_NMicro_email&utm_medium=ejP_NMicro_email&utm_campaign=ejp_NMicro for more information about our career opportunities. If you have any questions please click [here](mailto:editorial.publishing.jobs@springernature.com).**

Editorial comments:

[T]he reviewers raise a number of important points related to ensuring the robustness of your results with additional analysis. Furthermore, while the reviewers do suggest shortening the paper, we do NOT feel that the work would fulfill our editorial criteria for a Brief Communication, so we would not encourage you to shorten the manuscript to that format, but rather to hone in on the more interesting discoveries and expound upon those. We would be open to considering this paper as a Resource, rather than an Article.

Thank you once more for considering our study and for soliciting critical and constructive reviews. We have now revised the manuscript to address the editorial and reviewers' comments, as detailed below. The main changes over the previous version include:

- *We have added new analyses into patterns of prokaryotic diversity beyond species counts and rarefaction curves. These reveal that archaeal and bacterial clade size distributions (when including clades inferred from unbinned contigs, at the heart of our study) follow power laws, in line with century-old hypotheses by Willis & Yule. These 'Yule curves' and derived Yule coefficients vary by habitat and taxonomy, but are in line with estimates from reference taxonomies (like the GTDB) and for eukaryotic kingdoms, with interesting macroevolutionary implications for the organisation of biodiversity across the entire Tree of Life. These analyses are presented in (novel) Figure 4 and associated text.*
- *Newly added Extended Data Figures provide additional analyses and support existing analyses in the text, following reviewer feedback. In particular, we quantified noise and uncertainty in species count estimates and rarefactions (EDF 1 & 2), delved deeper into why some lineages remain 'unbinned' (EDF 3), studied gene cluster size distributions (EDF 4), present additional underlying 'Yule curves' supporting analyses shown in new Figure 4 (EDF 6 & 7), and provide more context on methodological choices (EDF 8 & 9).*
- *While new sections on Yule curves have been added (see above), we have streamlined the manuscript as per reviewer #2's suggestion by relegating some analyses to Extended Data Figures (Figure 2 -> EDF 5) and a Supplementary Discussion. Likewise, we have also moved more technical aspects of the Discussion section (regarding data processing caveats and sampling bias in the underlying public data) to the Supplement).*

With these revised and additional analyses we believe that our study fits the 'Article' category rather than a 'Resource'.

[T]he reviewers also raised a number of concerns about the code and its usability--this will need to be addressed in full before we can return the paper to reviewers (code should be fully accessible, documented with a README and able to be run with test datasets by the reviewers).

The original submission included barebones analysis code as a supplement and a more structured and documented version had been uploaded to <https://github.com/grp-schmidt/ms-census>. However, this was not clarified in a Code Availability statement in the original submission that was available to reviewers and only later communicated to the editorial office. We apologize for this inconvenience and for our oversight in the original submission. We have now further updated the deposited code to encompass also the newly added analyses and to add further documentation. In the revised submission, we have removed the originally submitted code supplement to avoid confusion, but have included the Code Availability statement with the correct link to GitHub.

Please include a data availability statement as a separate section after Methods but before references, under the heading "Data Availability". This section should inform readers about the availability of the data used to support the conclusions of your study. This information includes accession codes to public repositories (data banks for protein, DNA or RNA sequences, microarray, proteomics data etc...), references to source data published alongside the paper, unique identifiers such as URLs to data repository entries, or data set DOIs, and any other statement about data availability. At a minimum, you should include the following statement: "The data that support the findings of this study are available from the corresponding author upon request", mentioning any restrictions on availability. If DOIs are provided, we also strongly encourage including these in the Reference list (authors, title, publisher (repository name), identifier, year). For more guidance on how to write this section please see: <http://www.nature.com/authors/policies/data/data-availability-statements-data-citations.pdf>

Our study relies on re-processed publicly available data. Derived data underlying our presented analyses, such as Metagenome-Assembled Genomes, are available via the SPIRE resource as indicated in the text (spire.embl.de). We have now added a statement to clarify that extracted marker gene sequences from proGenomes3 and GTDB r220 genomes used for our analysis have also been deposited on spire.embl.de/downloads in combination with genes extracted from SPIRE assemblies. Marker gene phylogenies, including pre-generated visualizations of archaeal phylogenetic trees, were uploaded to the EBI Biostudies repository. However, due to a problem during data release, some of these datasets had to be re-uploaded under additional accessions shortly after the initial submission; we communicated to the editorial office, but the manuscript sent out for review did not include the updated data availability statement. We apologize for this inconvenience. Preprocessed and derived data to run uploaded analysis code has been deposited on Zenodo (<https://doi.org/10.5281/zenodo.17482698>).

The revised submission now contains the updated data availability statement, including DOI references for each data package.

EXTENDED DATA FIGURES

The manuscript now includes 9 Extended Data Figures:

- *Former main figure 2 was converted into Extended Figure 5 following a suggestion by reviewer #2. The corresponding discussion in the main text was moved into a Supplementary Discussion.*
- *Former supplementary figure 1 was converted into Extended Figure 2. Supplementary Figures 1 & 2 (formerly 2 & 3) remain as part of the supplement.*
- *Extended Figures 1, 3, 4 and 6-9 were newly added to support analyses in the main text.*

Reviewer #1

Prasoodanan and colleagues present an analysis of microbial diversity from single copy marker genes on unbinned metagenomic contigs. They deem this diversity "discoverable", as these are taxa that have been sequenced, but not yet genomically characterized (i.e., binned into genomes from metagenomes). They estimate the degree of discoverable microbial diversity across different environments and at different taxonomic ranks. The paper is well-written and the figures are elegant.

We thank the reviewer for their constructive comments; as detailed below, we have addressed them in the revised manuscript.

However, I feel like the paper brings up some major questions that are left unanswered, and could be expanded to become more than a census. The limitation of the work as simply a census of discoverable microbial diversity is that the extent of discoverable microbial diversity is constantly in flux, as more samples are sequenced and better software tools for assembly and binning are developed and employed. Therefore, a census is simply a snapshot and will quickly become dated.

Following the reviewer's suggestion, along with below comments on lineage 'rarity' and similar points raised by other reviewers, we have now added a series of analyses that, in our opinion, broaden the scope of our study and provide an additional perspective. We studied clade size distributions, i.e. patterns of prokaryotic diversity beyond total counts and rarefaction curves, and found that they followed power laws in line with century-old hypotheses, with implications for prokaryotic (macro-)evolution (new Figure 4 and corresponding Results and Discussion sections). Moreover, we agree with the reviewer that our work relies on a data snapshot (as pointed out in the Discussion section), yet we feel that the analyses of habitat-dependent species discovery coefficients and of discoverable 'deeper' clades extend beyond a mere census. Accordingly, and in line with the reviewer's further comment below, we have now also changed the title to 'Hidden but discoverable diversity in the global microbiome'.

However, the work lends itself to coming very close to answering some generalizable and important questions:

1. The primary question I am left with reading this work is: how much of the divergence between discoverable taxa and genomically characterized taxa is due to random chance vs systematic issues in current approaches in metagenomic assembly and binning? Are MAGs simply a random subsample of all unbinned sequences, and as we sequence and assemble more they will 'catch-up'? Or, are there some taxa that we are consistently undersampling in MAGs or missing entirely, possibly for biological reasons? For example: large genome sizes, high microdiversity, very repetitive genomes, even DNA modifications, may play a role here. If the degree to which the divergence between discoverable and genomically characterized taxa is systematic can be quantified, that would be a very valuable contribution to the field. The authors should just caution that this divergence will be methodologically dependent, e.g., long reads, single cells, and better algorithms might capture taxa that are currently systematically missed.

This is an excellent point. Following the reviewer's suggestion (and a related suggestion by reviewer #2), we have analysed possible bias in binning as the ratio of 'unbinned' genes per binned gene across individual phyla, using marker gene phylogenies. These results are now included as Extended Figure 3, reproduced below for reference. Our main findings:

- *In both Archaea and Bacteria, the ratio of the median number of species-level clusters containing only genes from **unbinned** contigs and clusters with at least one MAG- or reference genome-derived gene was around 4. In other words, we observed ~4 'unbinned' species per each binned species, when viewed across marker genes.*
- *There was significant variability between phyla: clades without cultured representatives (i.e., MAG only) had higher ratios of unbinned 'species' than those including isolate genomes (Cohen's $d = 0.33$; Wilcoxon $p = 0.01$). Interestingly, the bacterial phylum Thermotogota had **more** binned than unbinned genes; at the other end of the spectrum, Desulfobacterota_C had an unbinned cluster ratio of ~16.*
- *This 'unbinned species ratio' was positively associated with genomic GC content and to a lesser degree genome size, but not linked to coding density or phylum size (i.e., total number of species in clade). In a combined linear model, only GC content remained as a relevant predictor ($p \sim 10^{-6}$, $R^2 = 0.15$). In other words, lineages with higher GC content (and to a lesser extent, with larger genomes) have lower MAG recall.*

We fully agree with the reviewer that improved methods, in particular long read sequencing, will alleviate at least some of this bias. We have now updated the respective sections in both Results and Discussion to discuss the above findings in this context.

Extended Data Figure 3. Unbidden genes are enriched among uncultivated phyla and associated with genomic GC content. (A) The ratio of tips (i.e., marker gene clusters) containing only unbidden genes to those containing genes from reference genomes or MAGs (y axis) is shown for subtrees of individual recognized phyla, sampled iteratively from full marker gene phylogenies (see Methods). This unbidden ratio was higher for phyla without (purple) compared to those with (green) cultured representatives (Cohen's $d = 0.33$; Wilcoxon $p = 0.01$). **(B)** Ratio of unbidden to binned markers per phylum (y axis) shown against average genomic GC content, genome size, coding density (all derived from GTDB reference genomes per phylum) and the estimated number of species per phylum. R^2 and p values are shown for individual linear regressions of unbidden ratios against each variable, as well as for a combined multiple regression in which only GC content stood out as significantly associated in an ANOVA.

2. At line 84, the authors suggest that the discoverable yet hidden taxa in this data could be part of the "rare biosphere". But, I think they are missing an analysis of this: are the discoverable taxa consistently rare? How does rarity, both within sample and between samples, influence a taxon's ability to have been genomically characterized or not? Are we missing just the rare things, or are there some taxa that are ubiquitous yet rarely abundant, or are there some that are even abundant but still being missed? An analysis of this would greatly increase the impact of the work.

We thank the reviewer for raising this important point. First off, we apologize for the lack of precision in the original phrasing: the 'rare biosphere' is often defined via two criteria that can be somewhat orthogonal, i.e. low prevalence (observation in only very few samples) or low abundance (at or below detection limits within samples). In relying on assemblies, we can detect lineages with low prevalence, but observations on low abundance are only implicit (assuming insufficient coverage for assembly in all but very few samples, i.e. underdetection instead of 'true absence'). Nevertheless, our observation of a high fraction of singleton gene clusters is generally in line with expectations for a large number of (lowly abundant and/or lowly prevalent) 'rare biosphere' lineages.

We have now added two additional lines of analysis that further support this. First, the newly added Extended Figure 4 (reproduced below for reference) shows cumulative distributions for marker gene clusters, i.e. the cumulative fraction of total genes observed (y axis) against the fraction of gene clusters accounted for (x axis), broken down by data source (marker genes extracted from proGenomes3 isolate genomes, SPIRE MAGs or from unbinned contigs) and observable sequence pool.

Extended Data Figure 4. Cumulative distributions of marker genes per marker gene cluster. The cumulative fraction of marker genes (y axis) contained in ranked marker gene clusters (x axis) is shown for individual marker genes (grey lines) and the median marker gene (emphasis). Emphasized rectangles indicate the part of the distribution corresponding to singleton marker gene clusters. The left panels show distributions for marker genes extracted from proGenomes3 reference genomes: the leftmost part (orange) corresponds to gene clusters also observed in SPIRE MAGs or unbinned contigs; the middle section (yellow) correspond to clusters observed in unbinned contigs, but not MAGs; the rightmost section (blue) are clusters not observed in metagenomic assemblies. The central panels show distributions for marker genes extracted from SPIRE MAGs: marker gene clusters that were also observed in proGenomes3 or GTDB r220 genomes (blue, left) or not (right, orange). The rightmost panels show distributions for marker genes extracted from unbinned contigs: gene clusters also observed in reference genome sets (blue, left); those observed in SPIRE MAGs, but not reference genome sets (yellow, central); and those observed exclusively in unbinned contigs.

Some key observations:

- 50-60% of marker genes from unbinned contigs mapped to marker gene clusters from reference genomes or medium+ quality MAGs (rightmost panels in the enclosed figure). This indicates that they likely represent incomplete or (less frequently) overly contaminated partial MAGs for lineages represented by reference genomes or medium+ quality MAGs, i.e. up to half of 'unbinned' marker genes could simply be from 'bad' MAGs.
- A further 20-25% of unbinned marker genes were in non-singleton clusters, with the largest 'unbinned' clusters containing a median of 245 (Archaea) and 609 (Bacteria) sequences. This indicates that indeed, there are highly prevalent lineages that consistently evade binning (or fail subsequent QC). However, we feel that a further characterization of these (e.g. by linking marker gene clusters) is beyond the scope of the present study.
- The remaining ~20% of unbinned marker genes were indeed in singleton clusters (highlighted 'boxes' in the enclosed figure), representing up to ~75% of total gene clusters in Bacteria (bottom right panel).
- At the same time, 40-50% of gene clusters originating from isolate genomes (proGenomes3, left panel) were **not** observable among SPIRE MAGs or unbinned contigs; the majority of these (~30% of total PG3 clusters) were likewise singletons.

The overall gene cluster size distributions and the congruent patterns of dominating singletons among both isolate and 'unbinned' gene clusters are in line with expectations for lineages representing a low prevalence (and implicitly, low abundance) 'rare biosphere'.

Second, the newly added analyses on clade size distributions (Yule curves, Figure 4, reproduced below for reference) point in a similar direction. The fraction of (predicted) singleton species-level groups in our data fully aligns with expectations under power laws and Yule-Simon curves, i.e. we did not find disproportionately more (or less) singleton species than expected. Indeed, the Yule coefficients ρ for Bacteria and Archaea were **lower** when considering our full dataset including unbinned contigs than for reference taxonomies (GBIF, GTDB) and a 16S-based survey (Microbe Atlas). This corresponds to a higher degree of 'preferential attachment' (or concentration) of species among large genera (and deeper clades), and conversely a lower fraction of small (singleton or doubleton) species. In other words, our SPIRE-derived clustering predicts a clade size distribution that is **less dominated by singletons** than the GTDB or GBIF. At the same time, inferred ρ values in our dataset were closer approximations of those for eukaryotic kingdoms (plants, animals, fungi) and viruses based on reference taxonomies, and they were highly consistent between Archaea and Bacteria. Taken together, this supports that most of the observed singletons constitute a real biological signal, rather than artifacts or noise, and that the corresponding lineages are often not captured by MAGs because they are rare (low prevalence and/or low abundance).

Figure 4. Prokaryotic diversity follows Willis' law and Yule curves. (A) Log-log plot of clade size (number of species per clade; x axis) against clade count (number of clades containing x species; y axis) for the example bacterial marker gene *rsmD* (16S rRNA guanine methyltransferase, [TIGR00095]). Clade-level groups were inferred based on relative evolutionary divergence (RED) cutoffs from a full phylogeny at each taxonomic level (see Methods). Grey lines indicate fit Yule-Simon curves; estimated Willis coefficient ω and Yule-Simon coefficient ρ are provided (see Methods). **(B)** Equivalent to (A), but showing the number of clades (y axis) containing x subclasses (x axis) of the subordinate taxonomic level, i.e. genera-within-families to classes-within-phylum. **(C)** Yule-Simon coefficients estimated for individual habitats, i.e. only considering genes assembled from metagenomes of a focal habitat. X axis corresponds to the number of samples per habitat. **(D)** Yule-Simon coefficients estimated within individual recognized phyla (see Methods). Dotted lines indicate ρ estimated for prokaryotic domains, eukaryotic kingdoms and Viruses based on reference taxonomies from the Global Biodiversity Information Facility (GBIF), the GTDB r226 and the Microbe Atlas Project 16S rRNA OTUs (see Methods). Corresponding Yule curves, analogous to panel A, are shown in Extended Figure 7.

3. The authors report a fascinating novel deep archaeal lineage that is also very abundant (they have many species in this lineage). Are they able to use the data to tell us more about this lineage? Where is it found? How abundant is it in these samples? For example, it seems like it is sometimes found in the human gut (is that true in Figure 3?)?! Are they able to identify samples which perhaps co-assembly of would result in a genome? This would demonstrate how characterizing discoverable taxa can help guide their eventual genomic characterization.

It was not our intention to claim that we pinpointed a novel phylum in an individual marker gene tree, and we apologize if this was not stated clearly in the text. Single gene phylogenies, including for the comparatively 'well-behaved' GTDB markers, can be affected by phylogenetic artifacts among others as a result of heterogeneous evolutionary rates. For example, Archaea comprise a diversity of genome-reduced symbionts belonging to the DPANN archaea which appear to have faster evolutionary rates. The phylogenetic models used in our analyses cannot adequately model and account for these differences which may result in the recovery of long branches that represent LBA artifacts rather than new species. It is also possible that some archaea encode paralogs for some of these marker gene families or that distant homologs from viruses or eukaryotes have been included (see below). These aspects are difficult to control for in the absence of genomic context and external validation data – in particular for the resolution of deep branches. Furthermore, using complex models of evolution for large datasets it is computationally extremely prohibitive. We therefore refrain from attempts at characterizing novel phyla based on individual single-gene phylogenies; to do this properly, actual genomes or at least phylogenies of multiple concatenated informative markers are required in combination with the use of models that can account for rate differences (which is not an option for vast datasets of unbinned contigs). Thus, this is beyond the scope of the present study, although we agree with the reviewer's suggestion that a co-assembly workflow (as employed by the recently published Bin Chicken tool, doi 10.1101/2024.11.24.625082) could be a viable approach. Rather, we would argue that while individual phylogenies are not perfect (in spite of several conservative filtering steps), effects should even out when viewed across multiple trees, so that we can estimate how many lineages there are, even if we cannot say which they are. We have now added more text to the Discussion to clarify this:

These are estimates across many marker gene phylogenies so that issues affecting individual gene trees (such as paralogs or uneven evolutionary rates, [...]) are expected to mostly even out. Nevertheless, we note that the proper characterization of novel lineages requires genomic context, or at the very least phylogenies of multiple concatenated informative markers [...], beyond the scope of the present study.

*Thus we did not include the discussion of the 'novel' lineage in Figure 2 (formerly Figure 3) lightly. While we observed deeply branching 'novel' lineages in similar positions across several archaeal trees (visualizations for all trees are available under ebi.ac.uk/biostudies/studies/S-BSS2111), these appeared to contain some eukaryotic paralogs on closer inspection in some cases. For the *rpoE* tree shown in the main text, most tips in the 'novel' clade did not have close BLAST hits against the NCBI nucleotide db, whereas some others hit eukaryotic assemblies (yeasts or red algae) or 'uncultivated archaeon'. In our opinion, some of these underlying genes represent (deep) novelty, but feel that a thorough characterization of these lineages is beyond the scope of our study.*

4. The authors compute rarefaction curves but they do not compute their error. Are they able to do bootstrapping to put some error bars on rarefaction curves (maybe for supplementary)?

Thank you for pointing this out. We did in fact compute the rarefaction curves using a bootstrapped approach, re-sampling 5 times at each rarefaction step for each tested marker gene. The rarefaction curves in the main display items are therefore 'smoothed' in two steps: (i) using the mean across 5 random samples at each rarefaction step; (ii) using the median across marker genes at each rarefaction step. The curves are not, in fact, further smoothed during plotting. However, we appreciate that the large number of underlying data points per rarefaction step ($5 * n$ of marker genes) was not obvious due to the omission of error bars which we did to avoid visual clutter.

We have now added a new Extended Figure 1 (included below for reference) that displays the coefficient of variation ('cv'; the standard deviation divided by the mean) across bootstraps at each rarefaction step for each marker gene, as well as the median cv across marker genes. For Bacteria, the cv drops to <10% between 100-1,000 samples included in the rarefaction, and to <5% at <10k samples; for Archaea, values are slightly higher, but follow a similar trend and reach <5% at around 10k samples. In other words, while uncertainty is high (~100% cv) at initial rarefaction steps (including just 10-20 samples out of the total 92k), it rapidly drops and estimates are highly confident when crossing ~10k samples. Moreover, variation is consistent between the different data sources, i.e. uncertainty/noise is comparable in rarefaction curves for reference databases (PG3, GTDB), MAGs and unbinned genes (including singleton gene clusters). We note, however, that the cv drop at the far end of the curve (80-90k samples) is a mathematical necessity (re-sampling 95% of the data will necessarily provide more similar results) and should not be interpreted as a true decrease in uncertainty.

Extended Data Figure 1. Uncertainty and noise in marker gene-based rarefaction curves. The coefficient of variation (cv; standard deviation divided by mean; y axis) across 5 random permutations is shown per rarefaction step (x axis) for the different tested data sources. Thin data series correspond to individual marker genes, the median across marker genes is emphasized. Dotted lines indicate reference levels of 1%, 5% and 10% cv. Note that the sharp drop in cv towards the right is a mathematical necessity as noise between permutations decreases once nearly all samples are included.

Can they then estimate the number of entirely undiscovered taxa (e.g., extrapolating out the curves, perhaps using an approach like breakaway) using both unbinned and binned taxa?

We refrained from extrapolation for several reasons. First, collector's curves are inherently an interpolation method and therefore bounded by bias in the underlying data. As we discuss in the text, although we use a comprehensive and diverse set of samples and are confident that our results are informative, the underlying data was not obtained from a specifically designed global survey to estimate species richness, but from a collection of independent datasets. Most ecologists would agree that extrapolation from rarefaction curves is only valid if there is a consistent underlying design, e.g. stratified or random quadrant sampling in field work.

Second, existing methods to extrapolate from collector's curves rely on abundance/frequency data, or at least make strong assumptions about underlying abundance distributions and community composition (see e.g. PMID:23431585 or PMID:27512033). As a consequence, extrapolation is sometimes used to estimate community richness (a different problem to the one addressed in our study), but this has also been questioned for microbial datasets (see e.g. PMID:38869966).

That said, the species discovery coefficients shown in Fig 1C and discussed in the Results section point in this direction: while not estimating total global diversity (i.e., extrapolating to putative endpoints of rarefaction curves), they provide an estimate of 'how the curve will continue', i.e. of the rate of future species discovery assuming continued sampling with consistent underlying bias.

5. I really would take issue with the title "beyond genome-centric approaches". I really think the authors should change this title, because they have not shown that these discoverable taxa are in fact "beyond" it, merely that they haven't yet been characterized. Eventually, these taxa will be characterized by genome-centric approaches, whether it is with Illumina, long-read, single cell, cultivation, etc. "Beyond" seems to incorrectly imply forever out of reach. I would instead suggest "yet unidentified by genome-centric approaches" or something similar.

We thank the reviewer for pointing this out. It was not at all our intention to imply that unbinned lineages are 'forever out of reach' of MAGs and isolates. We have now updated the title ('Hidden but discoverable diversity in the global microbiome'), also to better reflect the newly added analyses – 'beyond' a census.

6. Figure 1: I think it'd make more sense for the SPIRE (MAGS) category to be blue, not orange. Because this is a "genome" category, more like proGenomes3 and GTDB r220.

We indeed worked with such a colour scheme initially, but the general feedback during discussions was that colleagues not involved in the study found this less intuitive. Your suggestion is to visually group genome sets together; our colour scheme instead groups reference databases (blue hues) and "contributions from global survey beyond reference databases" (orange-yellow), with orange MAGs as a middle level between reference genomes and unbinned contigs.

Reviewer #1 (Remarks on code availability):

7. Please add the code to a GitHub repository associated with the work for easier reproduction.

I cannot reproduce this analysis with code in PDF format, but upon visual inspection it seems reasonable.

Beyond the 'bare bones' code included as a supplement, documented and structured analysis code had been uploaded to a dedicated repository (<https://github.com/grp-schmidt/ms-census>). However, we erroneously did not include a Code Availability statement in the original submission and only amended this in follow-up communication with the editorial office, but failed to ensure that this information was passed on to reviewers. We sincerely apologize for the oversight!

The revised manuscript now includes the correct pointer and the underlying GitHub repository has been further updated to include novel analyses and additional documentation. Pre-processed and derived data to run code are available via Zenodo (<https://doi.org/10.5281/zenodo.17482698>).

Reviewer #2 (Remarks to the Author):

In this manuscript, Prasoodanan and Maistrenko et al. use the previously developed SPIRE platform to assess how much prokaryotic novelty remains to be discovered in existing sequence databases. They conduct gene-level analyses of metagenomic assemblies, which enables detection of organisms that cannot be recovered via traditional binning approaches. The major conclusions are that many prokaryotic genomes remain to be discovered in existing datasets, and that environmental sequencing yields more novelty than additional sequencing of the gastrointestinal tract.

This manuscript addresses an interesting and important question in microbiology, and the analysis appears to be exceptionally robust. However, the manuscript is built around a single central question (albeit a worthwhile one) and some aspects lacked clarity. With some revisions, I believe this work would be of sufficiently broad interest to warrant publication in Nature Microbiology.

Sincerely,
Matt Olm, CU Boulder

Dear Dr Olm, thank you for your encouraging and constructive feedback. As detailed below, we have carefully addressed your comments and revised our manuscript accordingly, also by adding several new Extended Figures to support our analyses. We apologize once more about the erroneous figure and equation rendering in the original proofs, as well as our failure to communicate that additional analysis code had indeed been uploaded to GitHub. These have now been corrected.

We fully agree with your comment (and related points by other reviewers) that our manuscript was built around a single central question. While this is in our view not necessarily a bad thing, reviewer comments on species rarity and size distributions prompted us to reanalyse our data with a broader view to clade size distributions, and in a broader macroevolutionary context. We feel that our new findings on power laws and Yule curves (presented in new Figure 4) address questions that go far beyond collector's curves and a 'census', and therefore add an interesting and biologically relevant facet to our study.

Specific comments:

1) The in-line figures in the main PDF sent to reviewers are low resolution and difficult to read; Figure 4 is entirely unreadable. I was eventually able to view the figures using the separate figure PDF files, but the core equations did not render anywhere (e.g., lines 507 and 520), and I could not find a way to view them.

We apologize for this technical problem. For the revised submission, we have made sure that all equations are correctly rendered in the proofs and that all figures are submitted in high resolution. Figures included directly in the proof may be of lower resolution, but all individual figures (and extended figures) have been submitted separately in full resolution as well.

2) The central question of the manuscript is: How much additional prokaryotic diversity can be detected in metagenomic assemblies by examining genes rather than relying on genome bins? This is an interesting and well-addressed question. However, I believe the manuscript would benefit from a more concise format- perhaps a short-format article with two figures. Figures 1, 2, and 4 all convey the same main point (that most novelty is found outside the human gastrointestinal tract), and consolidating them could improve impact and clarity.

We have now reorganised the manuscript by relegating Figure 2 (incremental habitat-stratified rarefactions) and the corresponding discussion to an Extended Figure and the supplement. We have moreover introduced an additional line of analyses and arguments around clade size distributions and patterns (not just total counts) of prokaryotic diversity, in (new) Figure 4 and the corresponding text.

3) While I'm convinced the core methods are sound, I believe additional supplemental figures would help readers understand the approach. For example, visualizations of the “very robust linear fits” mentioned on line 458

We thank the reviewer for pointing these out. The former figure S1 (now Extended Figure 2) shows that resulting species estimates based on these linear fits are indeed very consistent, but it was only referred to from the main text (Results) and not the Methods section you pointed out. We have now added an Extended Figure 8 (reproduced below for reference) to show example linear fits (of the type $n_of_species \sim n_of_gene_clusters$) for 10 (randomly selected) marker genes each for Archaea and Bacteria. Please note that the dark green data series ('spire.species') was included for reference, as it is discussed in the methods and included in the analysis code; fitted coefficients from that series were **not** used in our study.

Extended Data Figure 8. Marker gene-specific parameters to estimate species counts. Plots show the number of discovered species (y axis) against the number of sampled marker gene sequence clusters (x axis), for 10 randomly picked marker genes each for Archaea and Bacteria. The underlying data was obtained from iterative rarefaction along logarithmic steps (see Methods) for four different data series: 'GTDB r220' refers to (clustered) marker genes originating from GTDB r220 reference genomes, and corresponding species definitions; 'pg3' are marker genes extracted from proGenomes3, and corresponding GTDB-based species classifications; 'spire.ani_cluster' uses SPIRE MAGs clustered at 95% ANI as 'species' clusters (i.e., analogous to GTDB definitions) and MAG-derived marker genes; 'spire.species' uses species-level GTDB classifications of SPIRE MAGs instead. The deviation of the 'spire.species' series at higher marker gene cluster counts is likely due to 'underclassification' (a significant fraction of SPIRE MAGs were not classifiable to species level against the GTDB reference). Therefore, in analyses shown in the main text, coefficients fitted from the 'spire.ani_cluster' series were used to estimate species counts from (unbinned) marker gene clusters.

Moreover, included below is a plot of the estimated standard error of the linear regressions; each dot corresponds to one marker gene. The plot shows that errors for 'spire.ani_cluster' (i.e., 95% ANI-based genome clusters; light green) are comparable or lower (in Archaea) to those for fits for reference databases (species-level classifications in proGenomes3 and GTDB).

the species discovery curves on line 471,

We apologize for the confusion; the 'species discovery curves' referred to here are simply the species collector's curves / rarefaction curves shown in main Figure 1, Extended Figure 4 (formerly Figure 2) and figures S2&S3. We have now added more explicit pointers.

and the fitted lines referenced on line 516 would all be useful.

These refer to the underlying fits for the **species discovery coefficients** (see below for reproduced equation and details). We have now included plots (for randomly selected marker genes) of the type 'species discovered per rarefaction step' from which the coefficients were fitted as Extended Figure 9, reproduced below for reference.

Extended Data Figure 9. Fitting species discovery coefficients. Plots show the underlying data to fit ‘species discovery coefficients’ α (see Methods) for three randomly picked marker genes each for Archaea and Bacteria. Based on rarefactions, the log-log plots show the number of added species per sample (i.e., number of additionally ‘discovered’ species, y axis) per rarefaction step (i.e., number of samples added to the survey, x axis). Species discovery coefficients were estimated from the corresponding regressions in log-log space.

And of course, the equations on lines 507 and 520 need to be made readable.

Apologies once more about these; we have now double-checked that all equations are rendered correctly in the revision. For reference, they are reproduced below (with excerpts from the manuscript text for context):

$$S = k * N^{-\gamma} \quad (1)$$

$$\alpha = 1 - \gamma \quad (2)$$

where S is the number of newly ‘discovered’ species per N samples added to the survey; k is a proportionality constant; and γ is a saturation coefficient. [...] For more intuitive interpretability, we calculated a species discovery coefficient α as in eq (2) and summarised values across marker genes within each domain-specific set as median values. Thus defined, α scales on $[-\infty, 1]$, although only mildly negative values are expected to be observed in practice.

The newly added analyses (Willis and Yule curve fits, see above) rely on the following equations (text from Methods section in the manuscript):

$$C = a * S^{-\omega} \quad (3)$$

where C is the number of higher clades (e.g., genus) containing S sub-clades (e.g., species); a is a proportionality constant; and ω is the scaling coefficients, referred to as Willis coefficient here for simplicity. In other words, (3) relates the frequency of a clade to its size, with positive ω values indicating that large clades (e.g., genera containing many species) are exponentially less common than small clades (e.g., genera containing few or only one species). Otherwise, given the similarity of eq (3) to eq (1) above, the interpretation of ω is closely related to that of the above-defined species discovery coefficient α .

$$f(C_S) = \rho * B(C_S, \rho + 1) \quad (4)$$

where $f(C_S)$ is the non-negative frequency of clades of size S (i.e., containing S sub-clades); B is the beta function; and $\rho > 0$ is the Yule-Simon shape parameter. The distribution results from a Yule process where newly arising subclades preferentially attach to larger existing clades, proportionally to the existing clades’ sizes, sometimes colloquially referred to as a “rich get richer” process. For sufficiently large counts C_S , the frequency $f(C_S)$ approximates a power law as in eq (3), with coefficient $\omega \sim \rho + 1$. Parameter ρ indicates the shape of the distribution and the strength of ‘preferential attachment’, with large ρ indicating a sharper drop in the distribution (less pronounced preferential attachment, distribution less tail-heavy), whereas smaller ρ indicate more tail-heavy distributions and stronger preferential attachment effects. In other words, ρ can be thought of as a “rich get richer” coefficient, with low ρ values indicating a stronger dominance of fewer large clades in the clade size distribution.

4) An investigation into why these genes aren't binned would strengthen the study. Are they low coverage? Do they fall into low-quality bins, or no bins? Do they exhibit higher microdiversity than binned genes? Have you tried manually examining a few representative genes to understand why they failed to bin? Also, it would be helpful to include data on the frequency of species detected by novel genes beyond simply counting singletons and non-singletons. Finally, you might consider exploring why certain frequently assembled-but-never-binned genes consistently evade binning.

We feel that an investigation of microdiversity at scale would be computationally challenging, as it requires read mapping and SNV calling on tens of millions of representative genes (several hundred thousand marker gene clusters each for 122 bacterial marker genes, plus Archaea) across ~90k metagenomes. However, following the reviewer's suggestion (and a related comment by reviewer #1), we have analysed possible bias in binning as the ratio of 'unbinned' genes per binned gene across individual phyla, using marker gene phylogenies. These results are now included as Extended Figure 3, reproduced below for reference. Our main findings:

- *In both Archaea and Bacteria, the median number of species-level clusters containing only genes from **unbinned** contigs per cluster with at least one MAG- or reference genome-derived gene was around 4. In other words, we observed ~4 'unbinned' species per each binned species, when viewed across marker genes.*
- *There was significant variability between phyla: clades without cultured representatives (i.e., MAG only) had higher ratios of unbinned 'species' than those including isolate genomes (Cohen's $d = 0.33$; Wilcoxon $p = 0.01$). Interestingly, the bacterial phylum Thermotogota even had **more** binned than unbinned genes; at the other end of the spectrum, Desulfobacterota_C had an unbinned cluster ratio of ~16.*
- *This 'unbinned species ratio' was positively associated with genomic GC content and to a lesser degree genome size, but not linked to coding density or phylum size (i.e., total number of species in clade). In a combined linear model, only GC content remained as a relevant predictor ($p \sim 10^{-6}$, $R^2 = 0.15$). In other words, lineages with higher GC content (and to a lesser extent, with larger genomes) have lower MAG recall.*

Moreover, following the reviewer's suggestion, we have now analysed marker gene cluster size distributions, stratified by data source (marker genes from reference genomes, MAGs or unbinned contigs) and cluster context (i.e., marker gene clusters originating from reference genomes, MAGs or unbinned contigs). These results are now included as Extended Figure 4, also reproduced below for reference. Major observations:

- *50-60% of marker genes from unbinned contigs mapped to marker gene clusters from reference genomes or medium+ quality MAGs (rightmost panels in the enclosed figure). This indicates that they likely represent incomplete (or, less frequently) overly contaminated partial MAGs for lineages represented by existing genomes, i.e. up to half of unbinned marker genes could simply be from 'bad' MAGs that fail QC.*
- *A further 20-25% of unbinned marker genes were in non-singleton clusters, with the largest 'unbinned' clusters containing a median of 245 (Archaea) and 609 (Bacteria) sequences. This indicates that indeed, there are highly prevalent lineages that consistently evade binning (or fail subsequent QC). However, we feel that a further characterization of these (e.g. by linking marker gene clusters) is beyond the scope of the present study.*
- *The remaining ~20% of unbinned marker genes were indeed in singleton clusters (highlighted 'boxes' in the enclosed figure), representing up to ~75% of total gene clusters in Bacteria (bottom right panel).*

Extended Data Figure 3. Unbinned genes are enriched among uncultivated phyla and associated with genomic GC content. (A) The ratio of tips (i.e., marker gene clusters) containing only unbinned genes to those containing genes from reference genomes or MAGs (y axis) is shown for subtrees of individual recognized phyla, sampled iteratively from full marker gene phylogenies (see Methods). This unbinned ratio was higher for phyla without (purple) compared to those with (green) cultured representatives (Cohen's $d = 0.33$; Wilcoxon $p = 0.01$). (B) Ratio of unbinned to binned markers per phylum (y axis) shown against average genomic GC content, genome size, coding density (all derived from GTDB reference genomes per phylum) and the estimated number of species per phylum. R^2 and p values are shown for individual linear regressions of unbinned ratios against each variable, as well as for a combined multiple regression in which only GC content stood out as significantly associated in an ANOVA.

Extended Data Figure 4. Cumulative distributions of marker genes per marker gene cluster. The cumulative fraction of marker genes (y axis) contained in ranked marker gene clusters (x axis) is shown for individual marker genes (grey lines) and the median marker gene (emphasis). Emphasized rectangles indicate the part of the distribution corresponding to singleton marker gene clusters. The left panels show distributions for marker genes extracted from proGenomes3 reference genomes: the leftmost part (orange) corresponds to gene clusters also observed in SPIRE MAGs or unbinned contigs; the middle section (yellow) correspond to clusters observed in unbinned contigs, but not MAGs; the rightmost section (blue) are clusters not observed in metagenomic assemblies. The central panels show distributions for marker genes extracted from SPIRE MAGs: marker gene clusters that were also observed in proGenomes3 or GTDB r220 genomes (blue, left) or not (right, orange). The rightmost panels show distributions for marker genes extracted from unbinned contigs: gene clusters also observed in reference genome sets (blue, left); those observed in SPIRE MAGs, but not reference genome sets (yellow, central); and those observed exclusively in unbinned contigs.

Taken together, our data suggests that (i) most unbinned marker genes are part of low quality MAGs of genomically represented lineages (from isolates or MAGs); (ii) a significant fraction of marker gene clusters represent unbinned lineages that are prevalent but not (yet) genomically represented which (iii) may be due to genomic GC content and varies across phyla, with uncultivated phyla harboring higher fractions of unbinned lineages.

5) A comparison to singleM or sandpiper would be valuable. A quantitative comparison would be ideal (especially if you analyzed overlapping datasets), but even a brief discussion would suffice.

We fully agree that singleM & sandpiper are a relevant reference point for our study. We have updated previous pointers in the Introduction and Discussion sections to refer to the recent publication (at the time of original submission, only the preprint was available and therefore cited) and have made the context more explicit (tool names were not previously included), but also discuss differences in approaches and findings.

6) Why was RNA polymerase E chosen for Figure 3? It seems like a somewhat arbitrary choice.

The tree for rpoE was chosen purely as a representative example. Visualizations for the phylogenies of all other tested archaeal marker genes, as well as the phylogenies (Newick files) themselves and associated tip-level metadata were uploaded to the EBI BioStudies repository under accessions S-BSST2111, S-BSST2112, S-BSST2113, S-BSST2116 and S-BSST2117, although the originally submitted manuscript erroneously did not list all of them (some data had to be re-uploaded under new accessions). All other analyses described in the section pertain to analyses across marker genes.

7) I appreciated the inclusion of the caveats described on lines 257-262 and shown in Table 1. This is an important and well-executed control.

Thank you. This comment pertains to the discussion of ‘oversplitting’ GTDB phyla based on our automated RED cutoffs. The newly added Figure 4 (Yule curves) and particularly Extended Figure 6 (reproduced below for reference), provide further context, as they indicate that archaeal and bacterial classes and orders have significantly different clade size distributions, implying that they may sit at different ‘depths’ in the phylogeny.

Extended Data Figure 6. Willis and Yule-Simon coefficients across marker genes and taxonomic levels. *Estimated Willis coefficients ω (A) and Yule-Simon coefficients ρ (B) across taxonomic level. Each dot represents an estimate for one individual archaeal (left series) or bacterial (right series) marker gene. Data is provided for both ‘species per clade’ (clade size distributions defined as the total number of species per clade, i.e. per phylum, class, etc) and ‘subclades per clade’ (i.e., genera per family, families per order, etc).*

8) Please consider adding species information to Table 1. This would help readers interpret Figures 1 and 2.

Thank you for this suggestion. We realize that the numbers were previously distributed around the study a bit (across main text and S tables). We have now added counts to Table 1 as per your suggestion.

9) As experts in the field, do you believe that the singleton genes represent genuine biological sequences or assembly artifacts? My instinct is that many are likely artifacts given the known limitations of assemblers and how distinct the singleton line appears. I'm not questioning the accuracy of the manuscript's text, but I got the impression that the authors may believe most singleton sequences are real. This comes through in statements like "non-singletons represent a lower bound," whereas I might describe them as a best estimate since the same misassembly could feasibly occur twice.

This is a very interesting question and would frankly be worth an entire study in its own right. We applied relatively stringent cutoffs during marker gene detection and clustering, and even harsher cutoffs (e.g., $\geq 70\%$ alignment across informative positions) during phylogeny inference. So while there will certainly be spurious (mis-assembled) genes among the singleton gene clusters, we are indeed confident that the majority of singleton clusters represent true biological sequences. Our arguments are as follows:

- *While a recent benchmark estimated metagenomic misassembly errors to be in the range of $\sim 7\%$ of contigs (PMID:37126495), assembly errors are not randomly distributed and most are expected to be either (i) intra-species chimera (which would affect species-level clustering much less) or (ii) of such a type that HMMs would no longer detect a (partial) gene of sufficient length to pass our filters (PMID:36376928). In a large-scale benchmark of $\sim 300M$ assembled ORFs as part of the Global Microbial Gene Catalogue study (PMID:34912116), 75% of singleton genes were detectable in multiple metagenomic samples; 92% fell into known gene families; and only 0.4% were potential chimeras based on very inclusive criteria. We therefore estimate that misassembled genes with gene cluster-breaking errors are rare in our analysis.*
- *Our study does not explicitly account for alternative genetic codes; our ORF calls (and HMM searches) were conducted per metagenomic sample and gene callers perform poorly at detecting alternate codes (see e.g. PMID:38109938 or PMID:34751130), so it is likely that a significant number of real ORFs were prematurely truncated which would lead to an overall under-estimation of diversity.*
- *We have now added Extended Figure 4 (see above) which shows that the fraction of singleton clusters derived from MAGs and unbinned contigs is higher than for reference genomes (proGenomes3), but not disproportionately so. Indeed, a significant fraction of (singleton) clusters among reference genomes was also not detected among our metagenomic assemblies; we have slightly expanded the discussion of this point in the main text, also with a view to reviewer #1's comment regarding species rarity.*
- *In our newly included analyses on clade size distributions (Willis / Yule curves, Figure 4), the estimated 'rho' values (Yule-Simon shape parameter) for Archaea and Bacteria are **lower** for SPIRE-derived data (including unbinned contigs) than for GTDB or GBIF (Global Biodiversity Information Facility) archaeal and bacterial reference taxonomies, as well as than for 16S-based OTUs from the Microbe Atlas Project. Among other things, lower 'rho' values correspond to size distributions that are **less dominated by singletons** and doubletons. In other words, SPIRE-derived clade size distributions are broadly in line with reference databases, but if anything, they are less heavy on singleton and small clades.*

We have now included a condensed version of the above arguments as additional Supplementary Discussion.

Suggestion (non-essential):

Several figures could benefit from streamlining. In Figure 1A and related figures, I would suggest emphasizing your source descriptions on line 329 (cultivated organisms (proGenomes3), reference MAGs (GTDB r220) or dataset-specific MAGs (SPIRE)), rather than emphasizing the database names. Additionally, I recommend removing the “Unbinned all” category, since it’s less emphasized in the text and stretches the y-axis, making other differences harder to see. Finally, consider removing gridlines from several plots to make the visuals feel cleaner and less cluttered.

We have now re-organised data presentation in several ways. As indicated above, we have relegated Figure 2 and the corresponding discussion to the supplement. We have moreover added new Extended Figures that provide additional context on noise/uncertainty in the curves shown in Figure 1 (Extended Figure 1), on why (and which) lineages tend to remain unbinned (Extended Figure 3) and on marker gene cluster sizes (Extended Figure 4). As argued above, we believe our data and newly added analyses support that the majority of unbinned genes maps to existing lineages and that the long tail of singletons (‘unbinned all’) mostly represent a real biological signal and not noise; we have therefore opted to retain this category in the analyses.

Reviewer #2 (Remarks on code availability):

Kudos to the authors for making their code available- this is an important step toward transparency and reproducibility. However, the code is provided as a single text large text file of R code with no README, usage instructions, or list of dependencies. While sharing the code in any form is far better than not sharing it at all, the current format would make it quite difficult for others to reproduce the analysis or build upon it (which may be acceptable, depending on the intended audience).

Beyond the ‘bare bones’ code included as a supplement, documented and structured analysis code had been uploaded to a dedicated repository (<https://github.com/grp-schmidt/ms-census>). However, we erroneously did not include a Code Availability statement in the original submission and only amended this in follow-up communication with the editorial office, but failed to ensure that this information was passed on to reviewers. We sincerely apologize for the oversight!

The revised manuscript now includes the correct pointer and the underlying GitHub repository has been further updated to include novel analyses and additional documentation. Pre-processed and derived data to run code are available via Zenodo (<https://doi.org/10.5281/zenodo.17482698>).

Reviewer #3 (Remarks to the Author):

General comments

This study has considerable value: not so much in the essentially preliminary analysis of undiscovered taxa, but in highlighting the issue and in providing access to a toolset that can be used by other researchers in a much more focussed manner.

We thank the reviewer for their encouraging feedback and constructive comments.

Minor comments

Line 60. I disagree that ‘surveys of prokaryote diversity increasingly rely on isolate and MAGS’. Literature surveys will show that, for better or worse, 16S surveys still dominate, but are increasingly supplemented with MAG phylogenetics (reflecting the burgeoning of newly available technologies). I would argue that isolate-dependent surveys continue to grow, but at a fairly slow linear rate.

We fully agree: 16S (and other amplicon) data continue to dominate public repositories, and more new 16S datasets than shotgun metagenomes are generated each year. Our point was more that the focus in estimating global prokaryotic diversity (from integrated datasets) has been shifting from 16S data (surveys based on full length 16S or individual V regions were conducted 10-15 years ago) towards isolate genomes and MAGs recently. We realise that the previous phrasing did not fully reflect this. We have extended the discussion of 16S survey caveats in a (new) Supplementary Discussion section and have revised the main text to clarify as follows:

“While 16S rRNA data continues to dominate in public repositories (e.g., the Microbe Atlas Project encompasses 1.7M amplicon samples (Rodrigues et al. 2025), integrated surveys of prokaryotic diversity therefore increasingly rely on isolate and metagenome-assembled genomes (MAGs) [...]”

Line 194 and on: The authors might make the point, when discussing the relatively low taxon discovery rate in human associated metagenome data that this is perhaps not surprising, since humans, despite their diversity and complexity, represent a single species. The authors make a good point about the comparative heterogeneity of other habitats (e.g., see lines 371-373).

*Thank you for pointing this out. We have amended the text around (former) ll. 194f to point out that the bias is indeed even stronger given that humans are just one host species. Indeed, the only other ‘single species’ host category in the data (porcine gut, i.e. *Sus scrofa*) followed a curve roughly similar to human gut, although comparability was limited given the lower n (porcine gut sample count was just 4% of human gut samples) coupled with implicit study effects (far fewer animal studies than human studies).*

Line 267: The authors might consider explaining how their estimates of novel phylum level clades have any validity, considering the very high standard deviations.

*We explicitly do **not** claim the discovery (and description) of novel phyla as this is in our opinion not possible based on imperfect phylogenies of individual marker genes (with no connection via genomes) and an automated RED cutting algorithm. Rather, we estimate the ‘phylum-level’ diversity contained in unbinned contigs – i.e., how many phyla we estimate to be discoverable from these assemblies with better binning methods and/or better/deeper sequencing. That said, the uncertainty on reported ‘novel’ phylum-level clades is indeed high. In our opinion, this is for two main reasons:*

- *Limitations of the data. Databases like the GTDB rely on species phylogenies, i.e. on concatenated alignments across multiple marker genes. In contrast, we use trees of individual marker genes as (for unbinned genes) we do not have information on shared genome/species membership between individual markers. Individual phylogenies will necessarily be noisier in their resolution of deeper nodes, as different evolutionary rates, but also paralog effects, do not ‘even out’ as they do in (concatenated) species phylogenies. Rather, uncertainty in our estimates reflects differences between the individual markers used.*
- *Method. We ‘cut’ phylogenies along RED estimates using an automated algorithm, but with RED cutoff ranges that were (manually) defined for concatenated species phylogenies by the GTDB team. We validated that these cutoffs provide a good approximation also for individual marker genes, by running simulations for archaeal markers (as outlined in the Methods section; for Bacteria, these would have been too computationally intense). To our knowledge, the GTDB team manually validates deep (phylum-level) nodes based on RED values and additional expert curation of the full species tree. While we did inspect and validate all our marker gene phylogenies (and derived statistics), we feel that manually assigning ‘phylum-level’ nodes would introduce bias for the purpose of our analyses. Furthermore, due to the large size of our datasets, we were limited in the choice of phylogenetic models used (i.e. we could not use complex models of evolution that can more accurately account for rate heterogeneity and are generally used to infer species trees using concatenated marker sets) further emphasizing that our RED estimates should be seen as approximation only.*

In several marker gene phylogenies, ‘novel’ (unclassified) phylum-level nodes represented just 1-2 tips (but gene clusters with multiple genes from independent sources). Yet larger subtrees were common across many markers (cf the archaeal phylogenies uploaded under <https://www.ebi.ac.uk/biostudies/studies/S-BSST2111>) which suggests that (deep) novelty is likely to be discovered.

Reviewer #3 (Remarks on code availability):

The Code

Comments from Dr Pedro Lebre (Senior Postdoctoral Fellow, University of Pretoria: pedro.lebre@up.ac.za)

I checked the code and the syntax seems perfectly fine – it's harder to assess how it actually works without having the raw datasets, as from the scripts it looks like the authors used a bunch of pre-loaded Rstudio objects, which don't really have any correlation with the data available to the reader. They might need to clarify in more detail where these initial objects originated from.

Dear Dr Lebre, we thank you for your feedback. Beyond the 'bare bones' code included as a supplement, documented and structured analysis code had been uploaded to a dedicated repository (<https://github.com/grp-schmidt/ms-census>). However, we erroneously did not include a Code Availability statement in the original submission and only amended this in follow-up communication with the editorial office, but failed to ensure that this information was passed on to reviewers. We sincerely apologize for the oversight!

The revised manuscript now includes the correct pointer and the underlying GitHub repository has been further updated to include novel analyses and additional documentation. Pre-processed and derived data to run code are available via Zenodo (<https://doi.org/10.5281/zenodo.17482698>).